# ANSWER MATCHING OUTPERFORMS MULTIPLE CHOICE FOR LANGUAGE MODEL EVALUATION

## ABSTRACT

Multiple choice benchmarks have long been the workhorse of language model evaluation because grading multiple choice is objective and easy to automate. However, we show multiple choice questions from popular benchmarks can often be answered without even seeing the question. These shortcuts arise from a fundamental limitation of discriminative evaluation not shared by evaluations of the model's free-form, generative answers. Until recently, there appeared to be no viable, scalable alternative to multiple choice—but, we show that this has changed. We consider generative evaluation via what we call *answer matching*: Give the candidate model the question without the options, have it generate a free-form response, then use a modern language model with the reference answer to determine if the response matches the reference. To compare the validity of different evaluation strategies, we measure agreement with human grading, by annotating responses to MMLU-Pro and GPQA-Diamond questions. We find answer matching using recent models–even small ones–achieves near-perfect agreement, in the range of inter-annotator agreement. In contrast, both multiple choice evaluation and using LLM-as-a-judge without reference answers aligns poorly with human grading. Improved evaluations via answer matching are not merely a conceptual concern—it reduces costs, and significantly changes model rankings. Multiple choice benchmarks that seem saturated start showing room for improvement when evaluated with answer matching. In light of these findings, we discuss how to move the evaluation ecosystem from multiple choice to answer matching.

## 1 INTRODUCTION

Large language models impress with their capacity to generate fluid, free-form responses. But exactly how good are they? Evaluating generative language models is challenging, as there is no straightforward way to grade their unconstrained text output. Evaluations by human experts are too slow and costly to meet the demands of a sprawling evaluation ecosystem.

Benchmarks try to avoid the hard problem of evaluating free-form responses altogether by moving to multiple choice questions. Grading the picked choice is fast, objective, and easy to automate. But multiple choice does not directly evaluate generative capabilities; picking one out of multiple choices is rather a discriminative problem. A recent scalable alternative to multiple choice is LLM-as-judge, where a strong judge model directly scores a candidate model's answer, or, more commonly, compares the answers provided by two models (Zheng, 2023). Although compelling as a direct means of generative evaluation, LLM-as-judge runs into numerous biases (Tan et al., 2024a; Wang et al., 2024b).

As a result, both, recent benchmark creation efforts (Wang et al., 2024c; Zhang et al., 2025), and even frontier model releases(Yang et al., 2025; Liu et al., 2024; Google, 2025; Team Gemma et al., 2024), continue to fall back to multiple choice evaluations. Recent work even attempts to automatically generate multiple choice questions using language models, either from scratch (Yu et al., 2024), or by converting open-ended questions (Zhang et al., 2025). It almost appears as though there is no viable, scalable alternative to multiple choice, except in a few, specialized domains like code or math. In this work, we revisit the problem of grading free-form responses. We summarize our contributions below.

**Demonstrating Discriminative Shortcuts in Multiple Choice Benchmarks**: We start from a lightweight formal discussion that makes this problem of *generative evaluation* more precise and delineates it from discriminative evaluation. Against this backdrop, we show why multiple choice fails to solve generative evaluation. The reason is that *discriminative shortcuts* arising from the

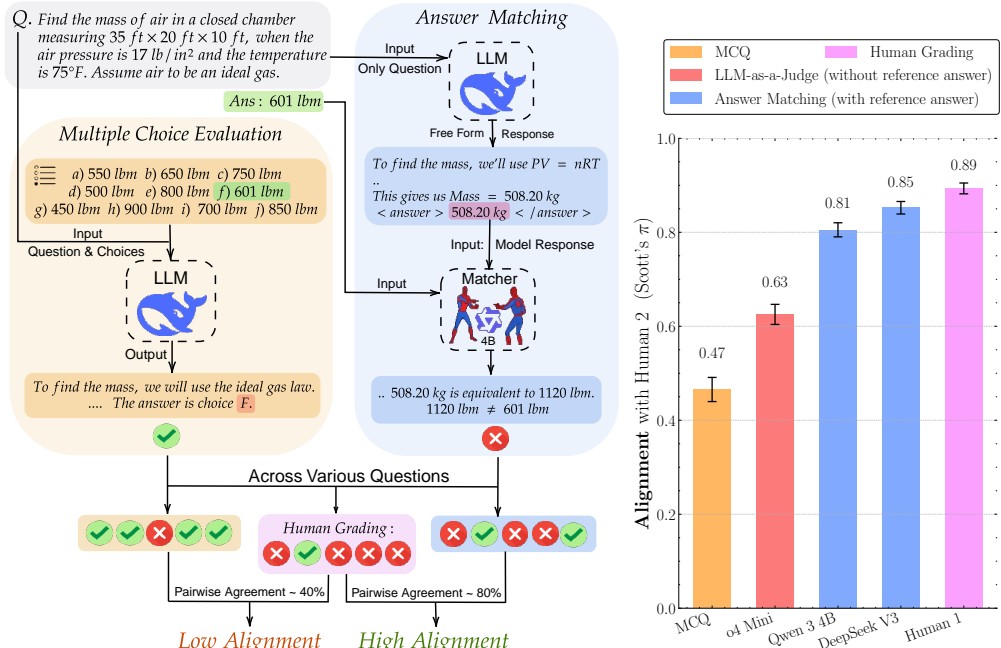

Figure 1: We show how multiple choice evaluations measure a discriminative task, rather than the generative capabilities language models are used for. (Left) On a multiple choice question, DeepSeek v3 picks the correct answer, perhaps due to **choice-only shortcuts like "odd one out"**, while giving the wrong response when prompted with just the question without choices. We propose using *Answer Matching*, which checks if a free-form model response matches the reference answer using a language model. (Right) On GPQA-Diamond, we find that answer matching aligns highly with carefully collected human annotations, significantly outperforming multiple choice. Even small language models as matchers can outperform frontier, expensive LLMs-as-a-Judge.

multiple choice format can sidestep generative evaluation. We demonstrate this fact with a simple experiment that reveals the propensity for shortcut learning in multiple choice benchmarks.

**Answer Matching Outperforms Multiple Choice Variants and LLM-as-Judge**: Our primary contribution, however, is to motivate a compelling, scalable means of generative evaluation, we call *answer matching*: Let the model generate a candidate answer given only the question. Then provide a second model with the correct answer and let this model decide whether the candidate answer *matches* the correct answer. At the outset, answer matching, also referred to as reference-guided grading, is a lesser known cousin of LLM-as-judge and it would seem to run into similar issues (Zheng, 2023; Zhu et al., 2024a). On the contrary, we find that answer matching, if done right, strongly outperforms all variants of multiple choice evaluations, and LLM-as-judge for generative evaluation.

**Comparing Evaluations by Annotating Model Responses on Popular Benchmarks**: To rigorously compare one evaluation method to another, we examine how well each method aligns with ground truth evaluations in three benchmarks: MATH (Hendrycks et al., 2021), MMLU-Pro (Wang et al., 2024c), and GPQA-Diamond (Rein et al., 2024). While answers to questions in MATH are automatically verifiable, answers to questions in MMLU-Pro and GPQA-Diamond are not. So we manually grade model responses and release our annotations publicly for further use.

**Demonstrating Recent Superiority of Answer Matching**: Based on our new annotations, we find that answer matching achieves alignment with ground truth evaluations that is vastly superior to the alternatives. Even relatively small judge models—when used for matching and not direct evaluation—achieve agreement rates close to the agreement between two humans. What's important here is that the matching model is *recent*. If you had evaluated answer matching two years ago, it would have fared a lot less convincingly. Answer matching has become a viable alternative only recently.

**Practical Implications for Benchmarking**: In principle, it is possible that even a flawed evaluation method can provide good model rankings. A method may fail to evaluate latent abilities on an absolute scale, but might still identify which of two given models is better. However, we demonstrate that the choice of evaluation method also *affects model rankings*. Using multiple choice will yield different rankings from those produced by answer matching. Further, benchmarks which seem close

to saturation in multiple choice format show more room for improvement in free-form generative evaluation using answer matching. Although LLM-Judge style evaluations are believed to be costly, we find that in practice the *cost of running a benchmark with answer matching is no more than that of multiple-choice evaluations*. We address potential reliability concerns and conclude with a discussion of how multiple choice datasets can be reused for generative evaluations with answer matching.

**Summary and outlook.** Answer matching is not new, but its superiority is. We argue that this qualitative change should inform future benchmark design. In principle, answer matching can be applied to any question from multiple-choice benchmarks, provided the question is specific enough that the correct choice—or a valid paraphrase—can be uniquely inferred. This can be achieved by filtering, as done in our human evaluation study, or rewriting the question and correct answer choice appropriately to utilize more samples. It might also be helpful to specifically design benchmarks for answer matching (Wei et al., 2024; 2025). For example, providing a reference list of multiple correct solutions for each question would be helpful.

Overall, the success of language models has recently been met with efforts to make harder multiple choice benchmarks. We show that multiple choice, by allowing discriminative shortcuts, is fundamentally easier than generating correct solutions. Rather than making multiple choice harder, the path forward may be to better align our evaluations with the generative capabilities we care about by leveraging the newfound capabilities of language models.

## 2 DISCRIMINATIVE SHORTCUTS TO MULTIPLE CHOICE EVALUATIONS

Whether answering a question or solving a task, generation can be formalized as the process of presenting the model $\mathcal{F}$ with a question $Q$, for which it generates a response $R = \mathcal{F}(Q)$, where $R \in \mathcal{S}$, the universe of all possible finite-length outputs. Let $\mathcal{A}_Q \subseteq \mathcal{S}$ be the set of correct answers for the question $Q$. We assume that the truth value of the response to a question can be determined, ensuring that the set $\mathcal{A}_Q$ is well-defined. Evaluating a generative model can therefore be formalized as the following decision problem—Is the generated response $R$ a member of the set of correct outputs $\mathcal{A}_Q$?

Unfortunately in natural language responses, a large number of paraphrases can convey the same point (Bhagat & Hovy, 2013) leading to $|\mathcal{A}_Q| > 1$. In these cases, it is hard to efficiently enumerate the set of correct answers, which in turn makes testing $R \in \mathcal{A}_Q$ challenging.

Figure 2: The correct answer set $\mathcal{A}_Q$ may contain paraphrases $a_1, a_2$. Generative evaluations thus involve testing the membership of response $R$ in $\mathcal{A}_Q$. Multiple choice evaluation, in contrast, tests whether a model can discriminate between a candidate answer $a_1$ and incorrect choices $\{w_1, w_2, \dots\}$

Circumventing this challenge is what popular question answering formats like multiple choice attempt to solve. Multiple choice evaluations give the model a question $Q$, and a list of choices consisting of a correct answer $a \in \mathcal{A}$ and $n$ incorrect choices $\{w_i\}_{i=1}^n \subset \mathcal{S} \setminus \mathcal{A}_Q$ called *distractors*. The model's response is $\hat{R} = \mathcal{F}(Q, \{a\} \cup \{w_i\}_Q)$, which is marked correct only if $\hat{R} = a$. In this way, the set of correct answers is now reduced to singleton—only $a$—enabling automatic grading. At first, it seems that multiple choice solves the problem of $|\mathcal{A}_Q| > 1$ we outlined above. However, on a closer look, changing the input from just $Q$ to $(Q, a, \{w_i\})$ fundamentally shifts the task from *generating* a correct response to *separating* the correct answer from the incorrect choices. The latter is traditionally considered a *discriminative* problem (see related work in Section 5).

**Experiment: Can multiple choice evaluations be answered without the question?** To demonstrate the extent to which multiple-choice benchmark can be solved discriminately in practice, we perform a simple experiment. We finetune a language model (Qwen3-4B) to predict the correct answer $a$ given only the choices $\{a\} \cup \{w_i\}$ without the question $Q$. For finetuning, we use the dedicated train split of the dataset whenever available; otherwise, we randomly split the test set 50-50, training on the first half and evaluating on the second half. Any accuracy obtained beyond chance in this way raises uncertainty about the extent to which accuracy on the dataset reflects generative question answering, as the model does not even know what question it is answering.

**Result 1: Yes, due to choice-only shortcuts.** Unfortunately, as shown in Figure 3, strikingly high accuracies can be achieved across popular datasets using *choice-only shortcuts*. We are not the

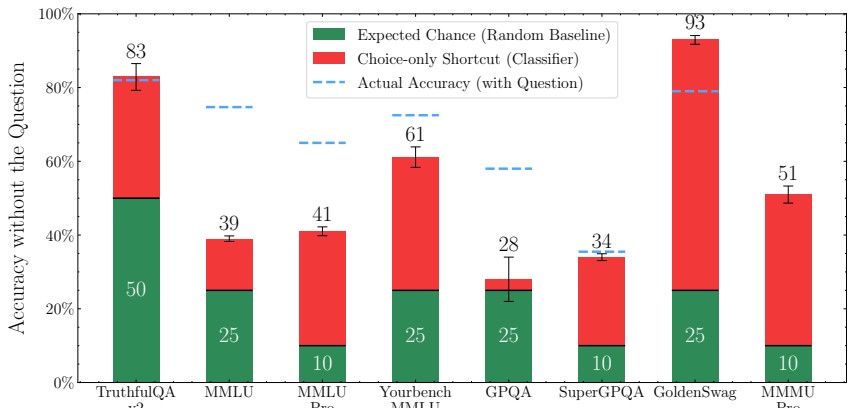

Figure 3: **Answering popular benchmarks without the question.** In red, we show the shortcut accuracy achieved by finetuning a discriminative classifier that sees only the answer choices beyond the random-guess baseline. The dashed blue line shows the model accuracy when prompted, without any finetuning. Strikingly, discrimination can provide high accuracies on popular benchmarks, attaining 83% on TruthfulQA-v2, 93% on GoldenSwag and 51% on MMMU-Pro (**without also the image**), showing the *statistical separability* of correct choices from incorrect ones.

first to point out this problem. For the popular TruthfulQA dataset (Lin et al., 2022), with four choices per question, Turner & Kurzeja (2025) show that identifying the "odd one out" can lead to large accuracies without looking at the question. This prompted the release of an updated version, TruthfulQA v2 (Evans et al., 2025), with only one incorrect choice. Once viewed from the lens of being a discriminative task, reducing the choices from four to two only makes it easier to exploit shortcuts! Indeed, we obtain an accuracy of 83% on Truthful QA v2, without even showing the question to the model. TruthfulQA is not special in being affected by this problem. Even on widely used hard, cross-domain benchmarks like MMLU, a non-trivial shortcut accuracy of 39% is seen, which might seem low considering chance accuracy is 25%, but is still interesting as it consists of questions from human examinations like GRE and USMLE.

**Result 2: LLM generated benchmarks are more prone to shortcuts and they are hard to remove.** It seems that the rising trend of using language-model-generated choices (Shashidhar et al., 2025) exacerbates the presence of choice-only shortcuts. For example, MMLU-Pro uses GPT4-Turbo to generate additional incorrect choices, increasing the number of choices from 4 to 10. However, compared to MMLU, this also increases our classifier's shortcut accuracy significantly, to 41% where chance is 10%. YourBench (Shashidhar et al., 2025) entirely generates the question and all choices from a document using an LLM, and on their "replication" of MMLU, we obtain a much higher shortcut accuracy (61%). Similarly, while GPQA (Rein et al., 2024) was designed with explicit measures to avoid choice-only shortcuts, on SuperGPQA, an LLM-assisted attempt to expand GPQA, we obtain much higher shortcut-accuracy relative to chance: $10\% \rightarrow 34\%$.

On older benchmarks like HellaSwag and ARC, prior work (Balepur et al., 2024; Li et al., 2021) has shown that choice-only prompting without the question achieves non-trivial accuracies. For example, Chizhov et al. (2025) show that on HellaSwag (Zellers et al., 2019), up to 70% shortcut accuracy is achievable by prompting without the question. Such validity issues in HellaSwag lead them to create a "corrected subset with substantially reduced effect of observed issues", which they call *GoldenSwag*. It has lower accuracy with choice-only prompting. However, there are myriad ways of exploiting inherent statistical separation between correct and incorrect choices, which are hard to remove manually. With choice-only finetuning, we find that the shortcut-accuracy on GoldenSwag is 93%, worse than HellaSwag where it is 87% (see Fig. 18 in Appendix). This exemplifies the difficulty of truly removing choice-only shortcuts from multiple choice datasets.

**Result 3: Answering "multimodal" benchmarks without an image or question.** Finally, our findings are not unique to language model benchmarks. On MMMU Pro (Yue et al., 2024), a visual question-answering benchmark with 10 choices, we obtain 51% shortcut-accuracy without showing the image or the question. We consider both the standard and vision subsets together. Thus, we find that discriminative shortcuts can plague multiple-choice benchmarks across domains.

**Discussion.** Our observations are not specific to any single language model. In fact, non-trivial shortcut accuracy can be achieved even by finetuning small embedding models like DeBERTa (see Appendix F). Viewing MCQs as a discriminative task explains recently raised concerns about how models can obtain non-trivial accuracy when prompted without the question (Balepur et al., 2024). For example, our backbone model for this experiment, Qwen3-4B, achieves a lower accuracy on MMLU-Pro (23%) when prompted with only choices, and not finetuned. However, it is not necessary that a language model always exploits such shortcuts when prompted with both the question and choices (Balepur & Rudinger, 2024). In this sense, our obtained shortcut accuracies *lower-bound* the fraction of samples which *can* be solved without the question. Our goal is not to catalog the extent to which shortcuts affect multiple choice datasets, for which curious readers can refer to prior works (Balepur et al., 2025). Rather, we demonstrate the discriminative nature of standard multiple choice evaluations is a fundamental way in which they diverge from the generative capabilities we set out to measure. We provide a conceptual discussion of the relative hardness of discriminative and generative tasks in Appendix B.

## 3 Answer Matching for Generative Evaluation

A simple way to prevent discriminative shortcuts is by not providing the model with choices in the input. In this section, we compare many evaluation methods of this form. What stands out as a compelling alternative is what we term *Answer matching*—where the model is simply tasked with providing a free-form response $R$, and then, another model checks whether the response $R$ matches with a provided reference answer $a$. Empirically we find that answer matching achieves alignment with ground truth evaluations that is vastly superior to all available alternatives. Even relatively small (but recent) grading models—when used for answer matching, not directly correctness assessment—achieve agreement rates comparable to the agreement between two human graders.

**Relation to LLM-as-Judge.** This kind of reference-guided scoring has occasionally been considered in the LLM-as-Judge literature (Thakur et al., 2024), but we argue that the distinction is crucial: LLM-as-Judge tasks a judge model $J$ with *verification*—given the question $Q$ and response $R$, it must decide whether $R$ is correct ($R \in \mathcal{A}_Q$). Traditionally (Zheng et al., 2023), the judge does not have access to a reference answer and has to assess the quality or correctness of a response, which leads to a host of issues documented in prior work (Tan et al., 2024b; Goel et al., 2025). In contrast, using a language model for answer matching only requires it to check if the model response is semantically or functionally equivalent to the reference answer in the context of the question, $R \equiv a$ given $Q$. Intuitively, matching seems easier than verifying the correctness of an arbitrary response (see Appendix B).

How then, do we determine what works best for generative evaluations? We propose collecting ground-truth evaluations of free-form model responses on popular benchmarks, and measure sample-level outcome alignment. Evaluations that are both scalable and yield outcomes more aligned with ground-truth assessments can be considered superior. We measure alignment using Scott's $\pi$, an inter-annotator agreement metric recommended in recent LLM-as-Judge literature (Thakur et al., 2024), for reasons elaborated in Appendix C.2.

### 3.1 Alignment on Verifiable Responses: MATH

We begin with the MATH dataset, where, as previously discussed, `MATH-Verify` library (Kydlicek et al., 2025) implements rule-based ground-truth evaluations of generative responses. Further, a parallel multiple-choice version is also available (Zhang et al., 2024). This allows us to compare the alignment of generative evaluations with multiple choice assessment on the same data distribution (see Figure 1 for illustration). In Figure 4 (right), we show that answer matching, even with the 1.7 billion parameter Qwen3 model (non-thinking mode), achieves near-perfect alignment with the ground-truth ($\pi = 0.97$). As for LLM-as-judge, even the much larger 671 billion parameter DeepSeek v3 model achieves only modest agreement $\pi = 0.72$, while as a matcher, it achieves $\pi = 0.98$.

**Could we fix MCQ in any other way?** Perhaps what stands out is that standard MCQ obtains only $\pi = 0.26$. This is mostly due to false positives ( 85% of errors) as reflected in the higher accuracy given by MCQ in Figure 4 (left), which is expected given that the task requires solving an easier discriminative problem. Next, we explore other variants of multiple choice evaluations that do not provide all choices in the input, thereby preventing discriminative shortcuts.

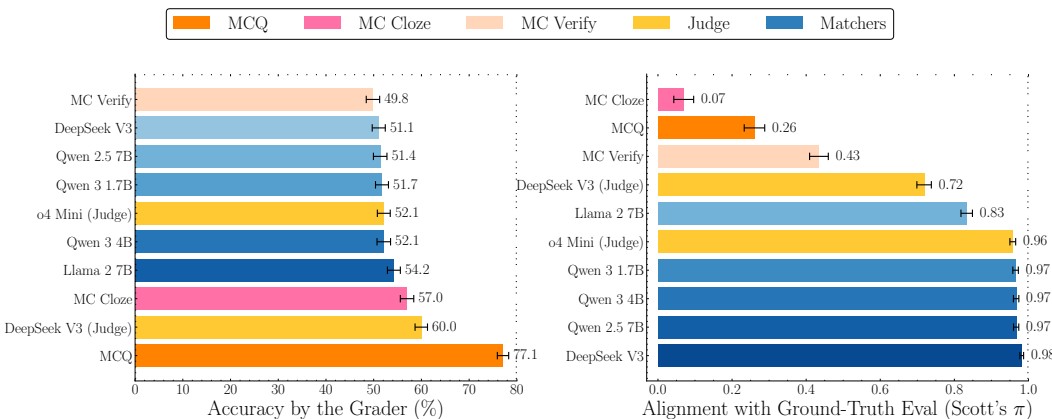

Figure 4: **Multiple choice inflates accuracy. Answer matching aligns highly with ground truth evaluation on MATH.** We evaluate the responses of Qwen2.5-7B on MATH Level 5 using different grading schemes. (Left) Each bar represents the accuracy estimated by a different evaluation. Classical MCQs inflate accuracy estimates. (Right) Bars rank graders from worst (top) to best (bottom). Even a small 1.7B matcher reaches Scott's $\pi = 0.97$, virtually indistinguishable from perfect agreement, whereas the MCQ evaluation aligns at only $\pi = 0.26$.

First, we consider *multiple choice verification* (Götting et al., 2025), where the model is given each choice for a question separately, and must independently determine whether it is the correct answer to the question. Formally, a model is considered correct in this setting if it outputs $\mathcal{F}(Q, a) =$ True and $\mathcal{F}(Q, w) =$ False for all $w \in \{w_i\}$. Many recently proposed multiple choice variants like including "None of the Above" (Elhady et al., 2025) or multiple correct choices essentially boil down to this verification task (Zhu et al., 2024b), as they force the model to evaluate the correctness of each choice independently. This grading method estimates similar accuracy (~50%) to the model evaluated as given by answer matching (~52%, see Fig. 4) suggesting it might lead to similar outcome as ground-truth evaluation. However, we find that its alignment is much poor ($\pi = 0.43$) than answer matching variants ($\pi = 0.97$) but better than providing all choices at once (MCQ, $\pi = 0.26$). In Appendix B, we discuss how verification is a strictly harder task than discrimination, and also discuss its hardness relation with the generative task, which has been of much recent interest (Swamy et al., 2025; Sinha et al., 2025).

Finally, *Multiple Choice Cloze* (Taylor, 1953) is a classical way to evaluate without allowing for choice discrimination. Although less popular now, it was, for example, the proposed format for the Abstract Reasoning Corpus (ARC) (Clark et al., 2018). It is implemented by only providing the model the question in the input, and then measuring completion likelihoods over all choices, picking the one assigned the highest likelihood. Unfortunately, it has even lower alignment than multiple choice, with its $\pi$ value (0.07) indicating its outcomes are almost independent from the ground-truth. This type of evaluation is entirely a non-generative likelihood evaluation, and so it is unclear how to fit in modern models which derive part of their prowess from generating a chain-of-thought before responding, potentially explaining its comparatively poor alignment.

## 3.2 ALIGNMENT ON NATURAL LANGUAGE RESPONSES: MMLU-PRO, GPQA-DIAMOND

Tasks like MATH (Hendrycks et al., 2021) have constrained numeric or latex expression as answers, which allow rule-based verification, and thus are not a use-case where one needs LMs to answer match. Do our observations generalize to benchmarks like MMLU-Pro and GPQA-Diamond which have natural language answers? To study this, we first create variants of these datasets which allow generative evaluation. We only provide the question to the model being evaluated, and use the correct choice as a reference answer for answer matching. Note that questions from these datasets often rely on the choices to convey the style and specificity of the intended answer. Thus, many of them are not unambiguously answerable in generative style, given just the question. Further, they often have multiple possible answers. This is only a ground-truth evaluation where sans semantic or functional equivalence, only one (set of) concept(s) is the correct answer. To ensure this, we *filter to questions which are 1) specific enough to be answered without choices, and 2) have a unique correct answer*.

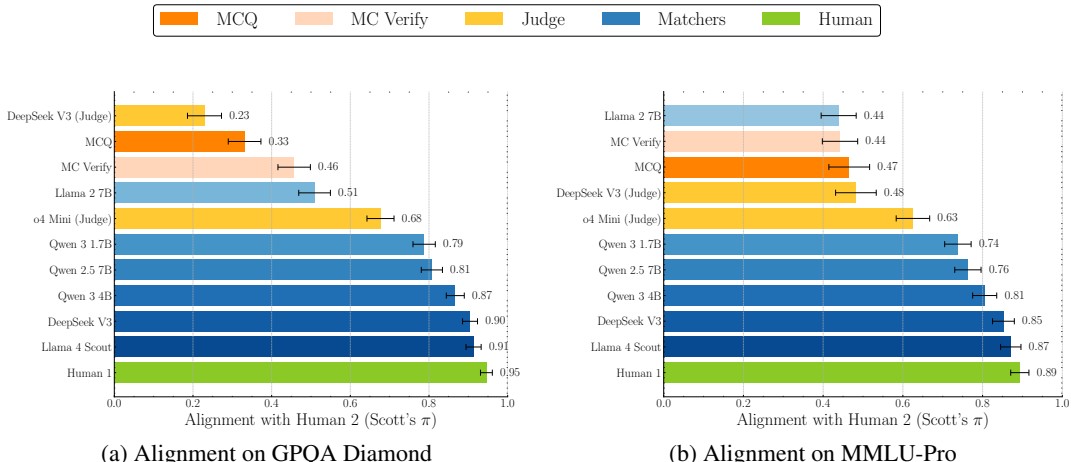

(a) Alignment on GPQA Diamond        (b) Alignment on MMLU-Pro

Figure 5: **Answer matching evaluations outperform multiple choice for MCQ benchmarks.** We compute alignment with human grading on GPQA-Diamond (left) and MMLU-Pro (right). Each panel plots alignment (Scott's $\pi$) between Human 2 and a range of automatic graders. Green bars (bottom) show inter-human alignment, with modern LLMs (blue) approaching human-level grading. Even small LMs like Qwen3-4B (without thinking) have high alignment, better than frontier LLMs-as-Judge. MCQ (and variants) are as, or more, poorly aligned with human judgement than Llama 2 7B.

Since there is no automated way to collect ground-truth here, we manually evaluate 800 model responses for correctness on both datasets, across four frontier models from different developers. Due to the cross-domain, knowledge intensive nature of these questions, a human can only grade by comparing responses to the reference answer. We then study how well different automatic evaluations align with human judgement. In our human study, we also prompt humans to rate whether the provided question and reference answer are specific enough, and can be arrived at uniquely (the question has a single correct answer). We report results on a subset of 493 questions on MMLU-Pro and 126 questions on GPQA-Diamond where both annotators think these properties are satisfied. We release the annotations and model outputs publicly for further use. Human graders extensively use tools like web search and calculators to make the annotations more accurate, spending more than a minute per response on average. Full setup details are provided in Appendix C.1.

**Modern LLMs are Human-Level Graders**. Figure 5 shows alignment with human judgements of MCQs, different LMs as judges, and LM matchers. Once again, we see a stark difference in alignment, with LM matchers consistently obtaining a higher value of Scott's $\pi$. We also perform an error analysis for LLM-as-judge, finding that for the frontier models (Deepseek V3, OpenAI o4-mini), errors disproportionately (80%+) arise from false positives–the judge finds responses correct which are marked incorrect in human annotation. We provide a detailed error analysis in Appendix E. This might be related to recent observations of sycophancy, an issue that also led to a rollback in ChatGPT[1]. Once again, it is striking that **small Qwen3 models have near-human level alignment, with the recent larger DeepSeek and Llama models having agreement within the range of inter-annotator disagreement**. In Appendix Appendix D.1, we also study the relationship between the size of a matcher and its performance for Qwen3 and Gemma3 model families and find that Qwen3-4B offers the sweet spot for being a strong yet cheap grader. Our findings are consistent with recent work (Krumdick et al., 2025) which shows that small language models provided with a reference answer perform better than larger models without a reference answer for grading in the domain of business and finance. Together, these findings confirm the validity of answer matching, showing it is now a viable alternative to scale evaluations at the frontier.

## 4   TOWARDS BENCHMARKING WITH ANSWER MATCHING

We now examine the implications of adopting answer matching within the benchmarking ecosystem, showing its impact on model rankings, benchmark saturation, and cost benefits. In Appendix A, we

---

[1]https://openai.com/index/expanding-on-sycophancy/

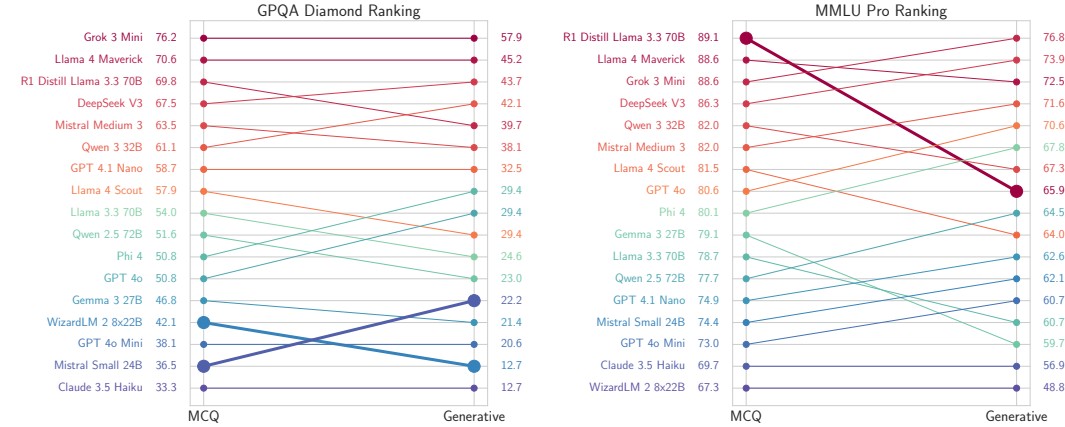

Figure 6: **Leaderboards change when moving from MCQ to answer-matching.** We evaluate generative responses to the *filtered subset* of GPQA-Diamond (Left) and MMLU-Pro (Right). Thick lines represent statistically significant changes based on the Compact Letter Display algorithm (Piepho, 2004). Proprietary models (GPT 4.1 Nano, 4o Mini, Claude 3.5 Haiku) climb on generative rankings, whereas some open-weight models (R1 Distill Llama 70B, WizardLM 2) drop markedly.

include both limitations of our study, such as the annotation process and not testing answer matching under optimization pressure, as well as limitations of answer matching, such as how it cannot be used on tasks with many semantically or functionally distinct correct answers.

**Rankings Change.** For public benchmarks, cardinal accuracy measurements and sample-wise alignment is perhaps of lesser importance than how models are ranked, as argued in Hardt (2025). After all, ultimately they serve as leaderboards that guide practitioners on what models to use. Does multiple choice—despite its issues—perhaps give the same model rankings as answer matching? Figure 6 shows that model rankings change quite a bit when directly measuring the more realistic generative use-case. For example, recent open-weights models trained via distillation like R1-Distill Llama 70B and Gemma-3 27B fall considerably on MMLU-Pro while Microsoft's WizardLM and Meta's Llama-4 Scout show large drops on GPQA Diamond. In contrast, we see proprietary models like GPT variants improve ranking in generative evaluation which seems plausible given that these models are typically optimized for chat-based applications. This might be a symptom of open-weight models being judged by the community on their performance in multiple choice benchmarks. If so, it highlights the criticality of the evaluation method in setting incentives for model selection.

**Answer Matching is cheaper than multiple choice.** A key concern in maintaining such public leaderboards is the potential cost of grading newly released models (Li et al., 2024). In Figure 7, we compare the *total cost* of evaluating 17 models, all that were shown in Figure 6, across both datasets" GPQA-Diamond and MMLU-Pro (see Appendix G for details). We find that answer matching, even using a frontier model (like DeepSeek v3), is no more expensive than multiple choice evaluations. Further, if using the much smaller Llama-4-Scout, which we found to have the best alignment with human grading, we observe a striking phenomenon: *the cost of answer matching can in fact be **lower** than that of multiple choice evaluations*. While this may seem counterintuitive, it is important to note that evaluation costs are primarily driven by the length of model responses. Using a language model matcher only incurs a small additional cost relative to the generation overhead.

We find that models generate longer responses for multiple choice than when they are asked to answer just the question (without choices). In the case of MCQs, models typically attempt to solve the question in a free-form manner first; if the answer they arrived at does not align with any of the given choices, they then try to reattempt the question or proceed to evaluate each choice individually, leading to longer response. *We observe this phenomenon across all the models we evaluated*, and provide detailed breakdown in Appendix G. Naturally, the evaluation cost can vary based on the model used for matching. Nonetheless, at the frontier, as inference-time compute is scaled, we expect that matching a response to a reference answer will require less compute than solving the task from scratch, as the former is easier. Thus, we believe the additional cost of answer matching will be marginal.

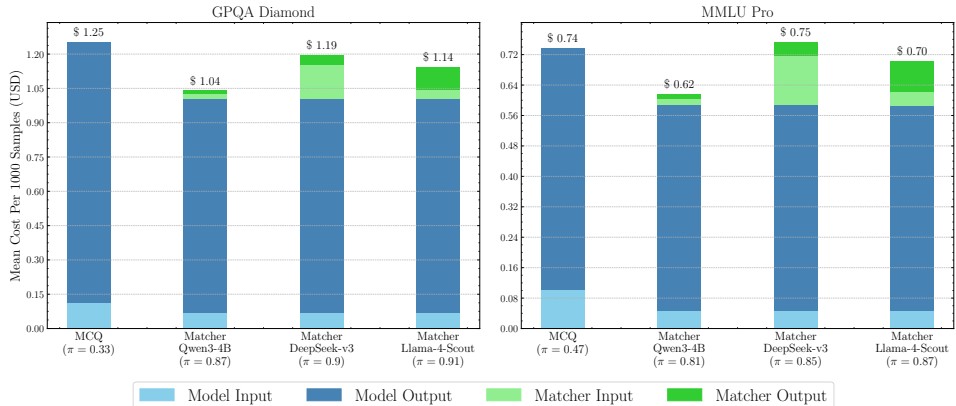

Figure 7: **Answer matching is cheaper than MCQ evaluation.** We provide a breakdown of evaluation cost averaged across 17 models. The cost of generating a response from a model, including its input and output tokens (blue bars), turns out to be significantly higher for multiple choice questions than free-form questions. Answer Matching (green) adds only a small overhead on top, even with frontier matchers (Llama-4-Scout, DeepSeek-v3) that have inter-human level alignment with human grading. Thus, answer matching not only improves evaluations, it also reduces costs.

**Reliability.** Another common concern with different methodologies for language model evaluations is their reliability. This concern has two primary aspects: reproducibility and robustness. First, for a long time, evaluations that rely on a language model as the grader were considered to have a reproducibility problem (Zhang et al., 2025), as only proprietary models, subject to depreciation, were sufficiently capable. However, this concern is now mitigated by both, progress in capabilities of open-weight models like DeepSeek-v3, and recent small models like Qwen3-4B being good at answer matching. To minimize variance, evaluations can be made deterministic (He & Lab, 2025).

As for robustness, we find that rankings remain highly stable even when using different models for answer matching. We show this with DeepSeek-v3, Llama-4-Scout, and Qwen3-4B in Appendix Figure 17. We also did not see any evidence of self-preference bias across these models, which is a significant issue in traditional LLM Judge setups (Wataoka et al., 2025). However, we do not test adversarial setups, where language models can be coerced to give favorable evaluations (Zheng et al., 2025; Geiping et al., 2024). Preliminary evidence suggests that such jailbreaks are getting harder to perform as models get more capable (Hughes et al., 2024; Panfilov et al., 2025). Until then, it might be useful to also report more adversarially robust evaluations like multiple choice alongside, so that high performance exclusively on LM-based answer matching evaluations can raise suspicion.

**Intrinsic Validity of Answer Matching is Recent.** One might also wonder, given that Llama 2 7B Chat (Touvron et al., 2023), released in July 2023, seems to match or beat the alignment of MCQ in our analysis, should we have moved on to LM answer matching much earlier? We argue that this is not the case.MCQ, while having poor construct validity as a measure of generative capabilities, is more reliable for what it claims to measure, namely, a model's multiple choice test performance. In contrast, older models lacked the intrinsic validity required for answer matching, as they performed poorly on this task. This has changed only recently, as newer models now achieve near-human agreement levels. We therefore believe that it is only with the recent generation of models that answer matching has clearly emerged as the superior mode of evaluation.

## 5 RELATED WORK

**Limitations of Multiple-Choice Evaluation.** Multiple-choice questions (MCQs) were introduced by Frederick J. Kelly in 1916 as a quick, objective, and scalable alternative to essay grading (Kelly, 1916). However, Kelly later warned that standardized tests built on MCQs reduce learning to mere finding shortcut solutions, leaving large gaps in testing answering ability. Over the past century, research in educational psychology has documented shortcomings of MCQ evaluations (Sampson & Boyer, 2001; Simkin & Kuechler, 2005; Farr et al., 1990; Roediger III & Marsh, 2005). Despite these drawbacks, MCQs still dominate large-scale testing — and, by extension, the evaluation of

language models. A long running critique of multiple-choice questions (MCQs) is that they primarily test the ability to *rank* (Haladyna et al., 2002; Ben-Simon et al., 1997) candidate choices or *validate* the correctness of a given choices (Haladyna & Downing, 1989) rather than to *generate* an answer from scratch (Ouyang et al., 2023; Bowman & Dahl, 2021; Balepur et al., 2025). Because the task is restricted to choosing among distractors, significant MCQ accuracy can be achieved just through shortcuts — e.g. relying on choice-only heuristics Turner & Kurzeja (2025); Balepur & Rudinger (2024) or inferring the intended question from the answer set (Balepur et al., 2024). This limitation is intrinsic to discriminative evaluation: the model is not tested on its ability to produce content beyond the provided choices. In contrast, answer-matching evaluations directly measure generative performance, on which models show lower accuracy (Myrzakhan et al., 2024).

**Generative Evaluation.** Answer-matching resembles classical Constructed Response Questions (CRQs) in educational testing: the model is tested on its ability to *generate* an answer. CRQs also span all levels of Bloom's taxonomy (Krathwohl, 2002), from recall to creation (Balepur et al., 2025). The main question to be tackled for automatic short-answer grading is *scoring* the generated response (Chen et al., 2019). Exact-string matching is too brittle; traditional n-gram metrics (BLEU, ROUGE, CIDEr) correlate only weakly with human judgments leading to other rule-based evaluations (Li et al., 2024). Subsequently, prior work has used alternatives like embedding-based similarity metrics (Bulian et al., 2022), including trained cross-encoders for grading (Risch et al., 2021). Recent work has proposed LLM-as-a-judge (Mañas et al., 2024; Zheng, 2023), which prompts LLMs to grade or critique model outputs, sometimes with access to grading rubrics or the answerer's chain of thought rationales (Ho et al., 2025). LLM-as-a-Judge evaluation without reference answers however has been often found to be brittle (Wang et al., 2024b; Goel et al., 2025), leading to uncertainty about the validity of LLM-based evaluation in general. In contrast, consistent with parallel work (Krumdick et al., 2025), we show that once LLMs are provided the reference answer, answer matching with recent LLMs can be a cheap way to score generative responses, that is better aligned with ground-truth evaluations. Note that prior work has already shown LLM-based answer matching works (Kamalloo et al., 2023; Wang et al., 2023), in that it has good correlation with human judgment for *open-domain* question answering. In fact, *recently* the community has already started using language models for evaluation via answer matching in frontier benchmarks like Humanity's Last Exam (Phan et al., 2025) and BrowserComp (Wei et al., 2025). Benchmarks like NovelQA (Wang et al., 2024a) use both multiple choice and answer matching evaluations for a specific task, but does not directly compare their validity. We show that not only is answer matching superior to multiple-choice but it is now also feasible to run reliably locally as small open-weight models achieve high alignment with human grading. Moreover, in the emerging regime of using test-time compute for harder tasks, we observe answer matching surprisingly incurs lower cost than multiple-choice evaluations.

## 6 CONCLUSION

In this work, we show that modern LLMs excel at matching free-form responses to reference answers. By carefully measuring alignment of different evaluation methods to ground-truth verification where available, and human grading, we find that such LLM-based *answer matching* is significantly more accurate at measuring generative capabilities than currently used alternatives, including many variants of multiple choice evaluation, and LLM-as-a-Judge without a reference answer. We demonstrate that this increase in validity also impacts the rankings of frontier models and also show that benchmarks which seem saturated show open up more room for improvement as models drop considerably in performance when required to generate free-form answers. Ultimately, what matters are the generative responses produced by language models, as these are what users interact with in practice. Scores on multiple choice benchmarks have long been questioned, but the lack of a scalable alternative has kept them popular in the community. The recent emergence of highly reliable LLM-based answer matching may represent a watershed moment—one that should inform the design of future benchmarks.

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

# Appendix

## CONTENTS

## A   LIMITATIONS AND CONSIDERATIONS

**Annotation Process.**   Some questions in MMLU-Pro and GPQA-Diamond require subject expertise to both check whether they are specific enough to be answered without choices, and also whether they have a unique answer. Further, there were disagreements when matching answers for even the filtered, shown in our alignment plots. While we are confident in the aggregate trends, individual annotations may be noisy. We release our annotations publicly and welcome community feedback to improve them.

**Gaming the Matcher.**   In this work, we did not study how robust are language models as answer matchers to gaming (for example, candidate outputting vague or multiple answers in the hope that it passes through the matcher) or optimization pressure. In the real-world, any evaluation scheme used will be optimized for, and given the ubiquity of LLM jailbreaks (Geiping et al., 2024), it is quite possible stronger models are needed for matching to rule out cheating models (Zheng et al., 2025; Hughes et al., 2024).

**On the hardness of matching.**   Relatedly, for some tasks, answer matching might be harder than simple verification. For example, in tasks with graph outputs, answer matching can require solving the graph isomorphism problem, whereas directly verifying the requisite graph properties can be much simpler.

**Answer matching can not always be used.**   For our alignment analysis, we filtered to questions with a unique correct answer (not counting paraphrases). This means our results do not apply to questions with multiple correct answers. In this case, either the dataset would have to provide as many semantically distinct valid answers as possible, or answer matching is no more guaranteed to provide correct evaluations. Moreover, the evaluation of many generative tasks can not be simply formulated with answer matching, e.g. translation, summarization, theorem proving, and coding. LLM judges with rubrics (Hashemi et al., 2024; Arora et al., 2025) or verification via execution (Chen et al., 2021) might be more suitable here.

# B  CONCEPTUAL FRAMEWORK

We now discuss the conceptual hardness relation between discrimination, verification, generation, and answer matching, followed by empirical results.

## B.1  ORACLE DEFINITIONS

For identifying hardness relations, akin to ones in complexity theory, we find it useful to consider oracles for each of these tasks, and then see when one oracle can be subsumed by another.

**Discrimination**: We define the discrimination oracle as a function that for a given question and set of choices, always picks the correct answer among the set of choices. Formally, $D^* = Q \times a \times \{w_i\}_Q \to a$, where $a$ is the correct answer and $w_i$ are not, i.e. $a \in \mathcal{A}_Q, w_i \notin \mathcal{A}_Q \ \forall w_i \in \{w_i\}_Q$.

**Verification**: We define the verification oracle as a function that for a given question and response $r$, checks if the response is a correct answer to the question. Formally, $V^* : Q \times r \to \{0, 1\}$, such that $V^*(Q, r) = 1$ if $r \in \mathcal{A}_Q, 0$ otherwise.

**Generation**: We define the generation oracle as a function that for a given question, outputs a correct answer. Formally, $G^* : Q \to a$, such that $a \in \mathcal{A}_Q$.

**Answer Matching**: We define the answer matching oracle as a function that for a given question, checks if a given response is semantically or functionally equivalent to a given reference answer, in the context of the question. Formally, $M^* : Q \times r \times a \to \{0, 1\}$, such that $M^*(Q, r, a) = 1$ if $r \equiv a|Q, 0$ otherwise.

## B.2  HARDNESS RELATIONS

If an oracle for a task $X$ can also be used as an oracle for task $Y$ in a setting, we can say that the hardness of task $X$ is at least that of task $Y$ for that setting, which we denote by $H(X) \geq H(Y)$.

**Verification can solve Discrimination but not vice-versa.**  First, we show that discrimination is strictly easier than verification. We can create a discrimination oracle $D*$ using a verification oracle $V*$, by applying the $V^*$ on each choice and outputting the one where the oracle returns $1$. On the other hand, discrimination as defined requires the guarantee that one of the provided choices is correct, and thus cannot be used for verification, which might only be given a wrong response.

There has been recent interest (Song et al., 2025) on the hardness relation between verification and generating a correct solution, with debate on how this varies across tasks (Swamy et al., 2025; Sinha et al., 2025).

**When there is a unique correct answer $|\mathcal{A}_Q| = 1$, Generation is computationally harder than Verification if $P \neq NP$.**  For $|\mathcal{A}_Q| = 1$, we can obtain a verification oracle by calling the generation oracle once with the question $Q$. We can simply check if the produced answer $a$ matches the response $r$ to be verified, as we assumed the correct answer is unique. However, to use a verification oracle as a generation oracle, we would have to enumerate strings in $\mathcal{S}$ and verify one by one until the verification oracle returns $1$. This can require exponential calls to the verification oracle in input length.

Note that even if a single answer had many semantically or functionally equivalent forms, one could obtain a verification oracle by also using a matching oracle to check if the output of the generation oracle is equivalent to the response to be verified.

**As the number of correct answers $|\mathcal{A}_Q|$ increases, Generation gets easier.**  Generation can sometimes be easier than verification as we only need to generate one correct answer, which can be easier than verification which requires distinguishing boundary correct and incorrect responses. Intuitively, think of it as throwing a dart inside the board gets easier as the board gets bigger, while precisely defining the board's boundary can be tough. More formally, generation gets easier as the fraction of correct answers among the universe of all possible strings increases, i.e. $\frac{|\mathcal{A}_Q|}{|\mathcal{S}|}$ gets larger. Consider the following randomized protcol which uses a verifier oracle to solve generation: Sample a

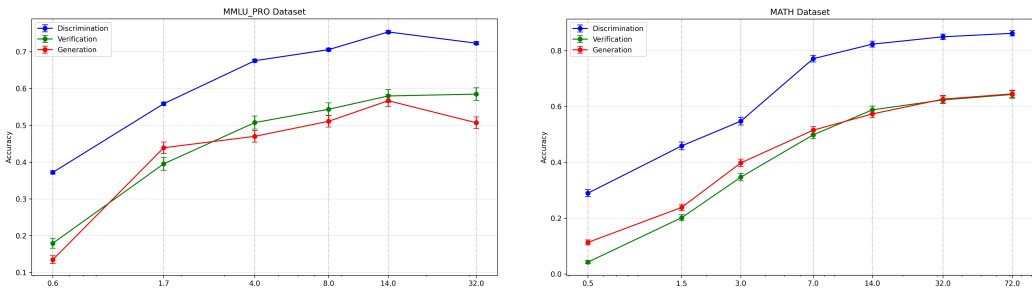

Figure 8: **Discrimination, Verification and Generation Performance on MMLU-Pro (L) and MATH (R)**: Discrimination accuracies are significantly higher. Verification and Generation accuracies are close together, with smaller models being worse at verification. While it can seem that models saturate both benchmarks when evaluated as MCQ, generative evals show there is still scope for much improvement.

string $r$ uniform randomly from the set of all strings $\mathcal{S}$. Apply the verifier oracle $V^*(Q, r)$. Repeat until the verifier returns $1$. This protocol has expected number of calls $\frac{|\mathcal{S}|}{|\mathcal{A}|}$ which could even be a constant if the set of correct answers is a constant (in input length) fraction of the total solution set. For example, consider the graph non-isomorphism problem, that is generating two graphs $G_1, G_2$ that are non-isomorphic. Here, generation can be simple, just output two graphs with a different number of edges. But verifying an arbitrary pair of graphs being non-isomorphic accurately is NP-Hard. However, note that verification does not always get harder as $|\mathcal{A}|$ gets larger, for example for the task "Generate an odd number", while $|\mathcal{A}|$ is large, verification can be done with a simple rule (mod 2) as we have a "clean decision boundary". Thus, whether generation is actually easier than verification depends on the complexity of verifying a solution for that particular task.

### B.3 EMPIRICAL OBSERVATIONS

We show how discrimination, verification and generation performance on MMLU Pro and MATH scales with model size.

**Setup.** On MMLU-Pro we plot Qwen3 thinking. On MATH (Level 5), we plot Qwen2.5 model series as Qwen3 saturates MATH even at 8B scale. For *discrimination*, we report the standard multiple choice accuracy with all choices given in the prompt to the model. For *verification*, we independently provide the model each option with the question and ask it to rate whether this being the answer is true or false. Only if the model marks just the correct option as true, and all the incorrect options as false, do we give mark it as correct for the question. We then measure accuracy as the mean verification correctness over all questions. For *generation* accuracy, we pose the question without any options, and use answer matching with DeepSeek v3 to check the correctness of the final model response (or ground truth in the case of MATH).

**Results.** Figure 8 shows empirical confirmation that discrimination (MCQ) is significantly easier than both verification and generation for models on popular benchmarks. From the MCQ results it can seem like even 7-8B saturated these two benchmarks. However, from the generative results, its clear that even the biggest 32B models achieve around 60% accuracy and the task itself is far from fully "solved" by these models.

**Why LLMs find judging without a reference answer harder than answer matching.** We see above that accurate verification can almost be as hard as generation on popular benchmarks. Note that verification is exactly the capability needed for obtaining an accurate LLM-as-a-judge without any reference answer. On the other hand, in answer matching, the model has access to the correct answer for grading. It can thus grade without knowing how to solve a question. In many tasks, answer matching can be as simple as checking if after unit conversions two values are equivalent. In natural language responses, models only need to be capable at detecting paraphrases. Note that there can

be tasks where answer matching requires great domain expertise and is quite hard. For example, it might be hard to verify if two proofs are equivalent. Checking if two graphs are structurally the same boils down to checking graph-isomorphism, which is NP-Hard. For example, consider the task of producing a graph with a pre-defined shortest path length between two nodes. There can be many such graphs, and answer matching can be hard to implement here for many reasons. First, one would have to provide all distinct graphs that satisfy this property as reference answers, and the answer matcher would have to check for a match against each of these for assigning correct outcomes. Second, finding the shortest path length between two nodes can be done with polynomial time algorithms, whereas checking if an output graph matches a reference graph is NP-Hard (graph isomorphism). In many more such cases, it is possible that verification is actually easier than answer matching, and thus LLM-as-a-judge without a reference answer works better.

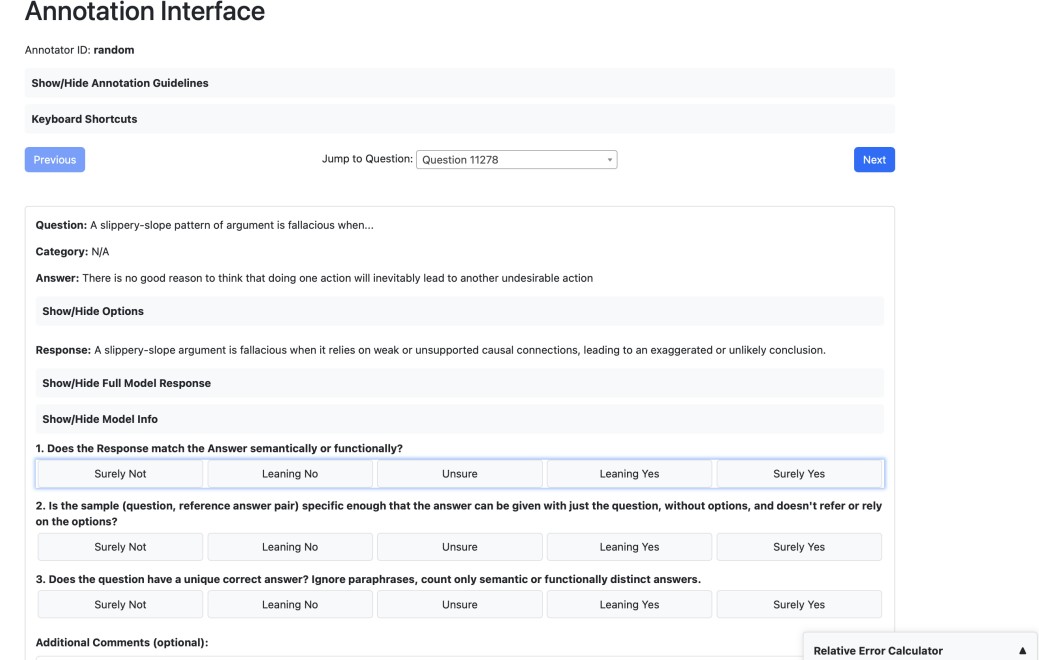

Figure 9: Screenshot of the annotation interface used by the authors to grade model responses with respect to the ground truth answer. The annotator has three primary tasks: 1) Match model response the answer; 2) Check whether the (question, reference answer) pair is specific enough to be answered without choices; 3) Check whether the question has multiple correct (semantically different) answers. For each task, the annotator is asked to a give a rating on a scale from 1 to 5 from No to Yes. This screenshot shows the interface for MMLU-Pro where human annotators had to grade only 1 response per question. For GPQA Diamond, additionally 3 more model responses were shown and for each of them, it was asked whether they match the answer semantically or not.

## C EXPERIMENTAL DETAILS

We now provide additional experimental details skipped in the main paper for brevity.

### C.1 HUMAN EVALUATION DESCRIPTION

Domains like math and code allow ground-truth verification using rule-base systems or unit tests but the same is not possible for benchmarks like MMLU-Pro and GPQA-Diamond which have natural language answers. As there is no automated way to collect ground-truth here, we did a human evaluation of models' responses by checking their free form response to the ground truth answer in the multiple choice variant. For each dataset, two of the authors independently did the annotation.

**Filtering MMLU-Pro for Human Study**: MMLU-Pro has 12,000 samples, and we could only carry out human grading on a smaller subset. We first use DeepSeek-V3-0324 to filter questions which cannot be answered without knowing the options, such as "Which of the following options is correct?". We provide a rubric-based scoring prompt following Myrzakhan et al. (2024, Table 1). The model provides ratings between 1 to 10 and then we use a high threshold of rating $\geq 8$ as this is enough to retain 5500 questions of MMLU-Pro. This filtering step skews the subject distribution of questions, as domains where questions have numeric answers (such as engineering, math, physics and chemistry) are more likely to be answerable without options in MMLU-Pro. So to obtain 800 questions for human annotation, we do a stratified sampling across subjects to obtain a balanced distribution across subjects. We then assign 200 questions each to four models: GPT-4o, DeepSeek-v3-0324, Llama 4 Maverick, and Qwen3-32B (with thinking).

For GQPA, we limit to the Diamond set of 198 questions which is reported as having high quality ground-truth labels. Since there are only 198 questions, we **do not filter** before the human evaluation, and evaluate the responses of the four models on every question, for a total of 792 human evaluations.

While evaluating responses on subset of both these datasets, we also mark whether each question is specific enough to be answerable in free form and whether the question has a unique answer. Ratings of these two aspects are later used to filter to the subset used for computing alignment between different grading schemes.

---

**Annotation guidelines for to human evaluators**

- For questions where what exactly to output (format / specificity etc.) seems unclear, use the "can question be answered in free-form" to mark it 'leaning no' or 'unsure'. Be strict with matching when the format/specificity is not same, only marking correct if model response is super-set of reference.

- For numerical answer questions, compute relative error by putting the numbers in the provided widget. (Relative error should not be more than $1\%$).

- When in doubt about whether an answer matches or not, or whether a different response is correct or not, use the options.

---

**Annotation Interface.** We use the annotation interface shown in Figure 9 for human ratings on the scale of 1 to 5 on multiple facets. Before starting annotation, we looked at some raw data from the datasets to understand how to classify questions suitable for answer matching. In this process, we developed a fixed set of rules for both annotators to follow (shown above).

For each question, we annotate whether it is specific enough to be answered without options (5), and whether it has a unique answer (5). For each model response, we annotate whether it is semantically equivalent to the provided reference answer or not. For our results, we use only questions that are rated specific ($\geq 4$) and unique ($\geq 4$), as here responses that match the reference answer ($\geq 4$) can be considered correct, and those that do not ($\leq 2$) can be considered wrong.

On MMLU Pro, each annotator spent around one minute on average per question (and response). We extensively use tools like web search and calculators to make the annotations more accurate. On GPQA Diamond, each annotator spent around five minutes per question as the questions are much more difficult, and the annotator was also required to grade four model responses simultaneously.

For GPQA-Diamond, as we annotated the whole dataset (due to its small size), we performed another quality check given it is the *diamond* (high-quality) subset of GPQA and that fact that our annotated version can be used in downstream evaluations. After individual annotation, annotators went through the questions where they disagreed upon (in terms of its specificity and answer uniqueness) to discuss the ambiguity and updated the annotation if they reached a common ground.

Post filtering, we were left with 493 questions of MMLU-Pro and 126 questions of GPQA-Diamond. Given that filtering questions for free-form answerable eliminates subjective questions, we plot the change in distribution of the question subjects in Figure 10 (MMLU Pro) and Figure 11 (GPQA-Diamond). For example, in MMLU Pro, we observe that the number of questions in law, psychology and history decrease significantly.

**Evaluating Thinking Models** Chain of thought prompting, and training models to use inference-time compute via thinking tokens is rising in popularity as it enables gains in performance on many reasoning tasks. To evaluate such models, we let them reason however they want but only ask them to give final answer inside XML tags, and subsequently evaluate only the final answer provided inside answer tags (see Fig 1 for an example).

For each model, we used maximum token limit of $16,384$. We used temperature of $0.6$ for thinking models and $0.3$ for non-thinking models.

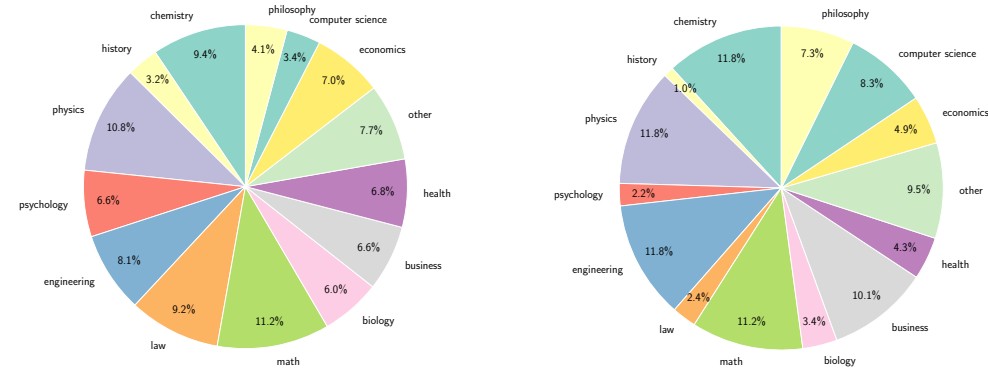

Figure 10: Change in subject distribution of MMLU Pro before (left) and after filtering (right) to questions which are suitable for answer matching evaluation.

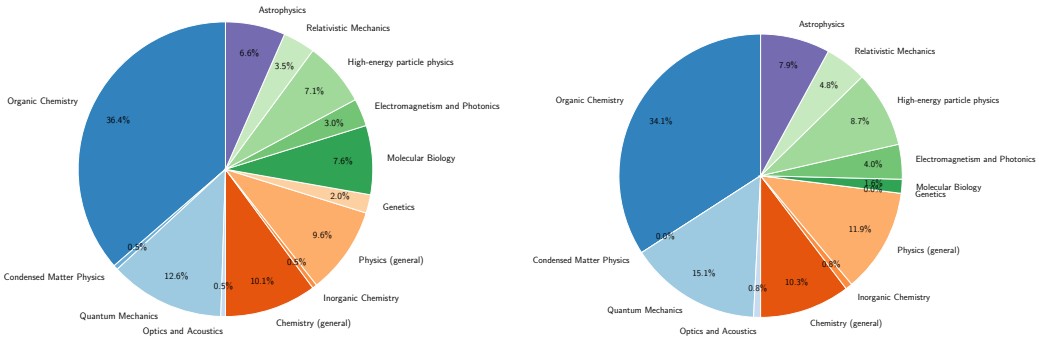

Figure 11: Change in subject distribution of GPQA Diamond before (left) and after (right) filtering to questions which are suitable for answer matching evaluation.

## C.2  ALIGNMENT WITH HUMAN EVALUATIONS ON MMLU-PRO, GPQA-DIAMOND

Naively computing the percentage of samples where the outcomes of the MCQ or matcher are the same as the human evaluator has two major issues. 1) The underlying distribution can be imbalanced if the accuracy of the models being evaluated is not 50%. This can allow no-information evaluators to score highly. For example, if the underlying model accuracy is 70%, always evaluating the model response as "correct" will give 70% agreement. 2) The agreement can be inflated by random guessing. Suppose the evaluator only knows how to evaluate 60% of the responses correctly. Since the underlying "correct" or "wrong" grading task is binary, by random guessing on the remaining 40% responses, agreement can be inflated to 80%.

We thus use Scott's $\pi$ to measure alignment of the evaluation with ground-truth, consistent with (Thakur et al., 2024). It measures observed agreement ($P_o$) in excess of what is expected by chance ($P_e$) assuming both raters arise from the same marginal distribution.

$$\pi = \frac{P_o - P_e}{1 - P_e}, P_e = (\frac{p+q}{2})^2 + (\frac{1-p+1-q}{2})^2$$

where say $p, q$ are probability assigned to "correct" by the two evaluations (one of which is ground-truth). Note that in our setting, Scott's $\pi$ offers a crucial benefit over the alternative annotator agreement metric, Cohen's $\kappa$. For a fixed observed agreement, Cohen's $\kappa$ becomes higher as the marginal assigned to "correct" and "incorrect" across all responses gets further away the marginals of the ground-truth evaluation. This is highly undesirable, and something $\pi$ avoids (Krippendorff, 2004).

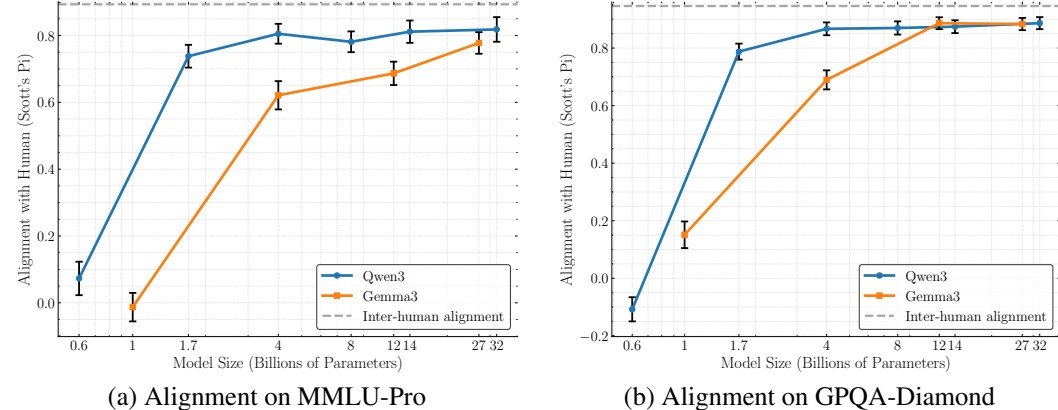

(a) Alignment on MMLU-Pro            (b) Alignment on GPQA-Diamond

Figure 12: **Alignment of matchers across varying scales.** We plot the alignment of matchers with humans (calculated via Scott's $\pi$) for models of varying parameters in the Qwen3 and Gemma3 family.

# D    ADDITIONAL RESULTS

## D.1    IMPACT OF MATCHER

To study the optimal trade-off between matcher model size and cost, we evaluate the alignment of models at different scales with humans. Specifically, we choose the Qwen3 and Gemma3 family as they are not only open-weight but also offer models of varying sizes (1B-32B) and measure their performance across MMLU-Pro and GPQA-Diamond benchmarks. Gemma3 models were release in March 2025 whereas Qwen3 models were released in April 2025. We evaluated the Qwen3 models in **non-thinking** mode.

We plot the performance and model size in Figure 12 and also report its their cost in Table 1. For the Qwen3 family, we find that on both MMLU Pro and GPQA Diamond, there is a rapid improvement from 1B to 4B , and after that the alignment plateaus, suggesting Qwen3 4B might be a sweet spot for cost-accuracy tradeoff. For the Gemma3 family, on GPQA Diamond we see alignment improve till 12B but not much after that, whereas for MMLU Pro 27B clearly is a better grader than 12B, perhaps because the latter is a more subjective dataset. This shows that the optimal choice of matcher size can be task dependent, and also depends on the cost-accuracy preferences of the user. That said, Qwen3 4B stands out as a cheap yet strong grader for both MMLU Pro and GPQA Diamond. Our analysis demonstrates that effective answer matching, depending on the task, may not require expensive frontier models—adequately-sized open-weight models (4B–8B) provide strong alignment at minimal cost.

## D.2    CHANGE IN RANKINGS

In Section 3.1, we showed the accuracy and alignment of different graders on MATH Level 5. Here, we also plot the same for GPQA-Diamond in Figure 15(a) and for MMLU-Pro in Figure 15(b). We find that MCQ estimates the highest accuracy followed by LLM-judges. These judges overestimate the performance as they often quickly conclude responses to be correct at surface level without engaging deeper. Meanwhile, language models as matcher give accuracy similar to humans. Even between humans, the accuracies are not same and we found this difference to be higher in MMLU Pro (compared to GPQA-Diamond) as the questions are more subjective.

**Ranking Changes.** In Section 4, we showed the change in rankings when shifting from MCQ to generative evaluations. There, we showed this on the subset which humans found suitable for answer matching. While filtering is essential to assess the true performance of a model (cardinal value) on the free-form version of the benchmark, it may not be important for comparing model rankings as questions which cannot be answered free-form can be considered label noise (from the perspective of answer-matching evaluation). Thus, we also plot the change in model rankings on the whole dataset for GPQA-Diamond and MMLU-Pro in Figure 16. We observe more significant changes in MMLU Pro with both Llama-4-Maverick and Qwen3-32B dropping considerably which was not the

Table 1: **Alignment of models of varying scale from the Qwen3 and Gemma3 family on the task of answer matching.** The Qwen3 models are evaluated in the **non-thinking** mode. Pricing is taken from OpenRouter.ai; models marked N/A are not served by its API but can be self-hosted. Scott's $\pi$ measures alignment with human graders.

| Parameters | Cost | MMLU-Pro | GPQA Diamond |
|---|---|---|---|
| **Qwen3 Models** | | **Scott's Pi** | |
| 0.6B | N/A | ~0.07 | ~0.1 |
| 1.7B | N/A | ~0.74 | ~0.79 |
| 4B | N/A | ~0.80 | ~0.87 |
| 8B | 0.14 | ~0.78 | ~0.87 |
| 14B | 0.22 | ~0.81 | ~0.87 |
| 32B | 0.2 | ~0.82 | ~0.88 |
| **Gemma3 Models** | | **Scott's Pi** | |
| 1B | N/A | ~-0.01 | ~0.15 |
| 4B | 0.07 | ~0.62 | ~0.69 |
| 9B | 0.10 | ~0.69 | ~0.89 |
| 27B | 0.16 | ~0.78 | ~0.88 |
| **Inter-human alignment** | | 0.89 | 0.95 |

case earlier. In GPQA-Diamond, we obtain similar conclusions as in our earlier observations from Figure 6 in Section 4 of the main paper.

As models are used for answer matching, it is also important to study their robustness. Thus, we investigate the change in rankings when different language models are used as matchers. In Figure 17, we plot the ranking changes across DeepSeek-v3, Llama-4-Scout and Qwen3-4B as matchers. We see that none of the changes are significant but the rankings are also not perfectly same. There are very few changes between Llama-4-Scout and DeepSeek-v3 while it is slightly higher between Qwen3-4B and DeepSeek-v3. In terms of benchmarking, it will be useful to fix a language model (say Llama-4-Scout, which is both cheap and has high alignment with human evaluation) as matcher for consistent reproduction in the community.

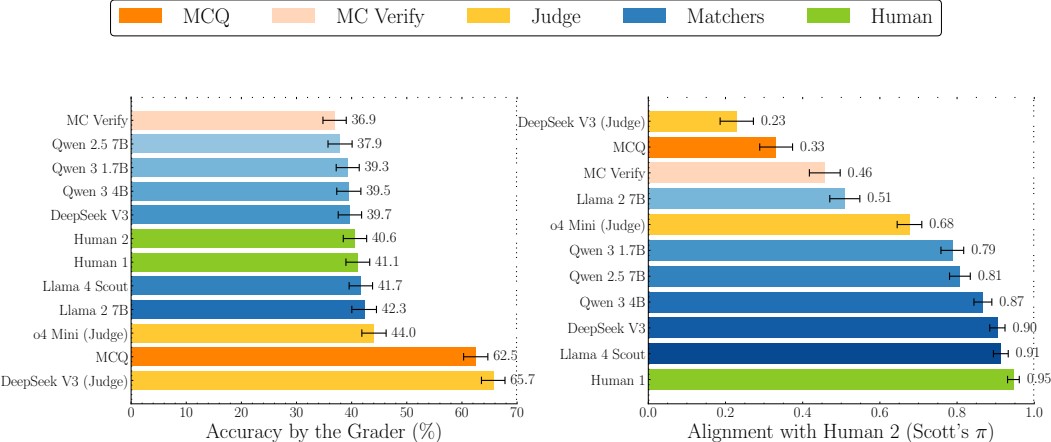

Figure 13: GPQA Diamond.

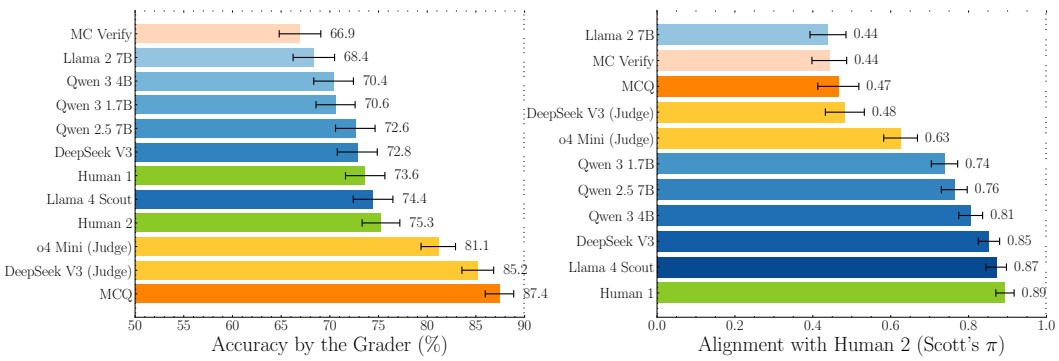

Figure 14: MMLU Pro.

Figure 15: Accuracy estimated by different graders and their alignment with human evaluation on two popular datasets we use in our study.

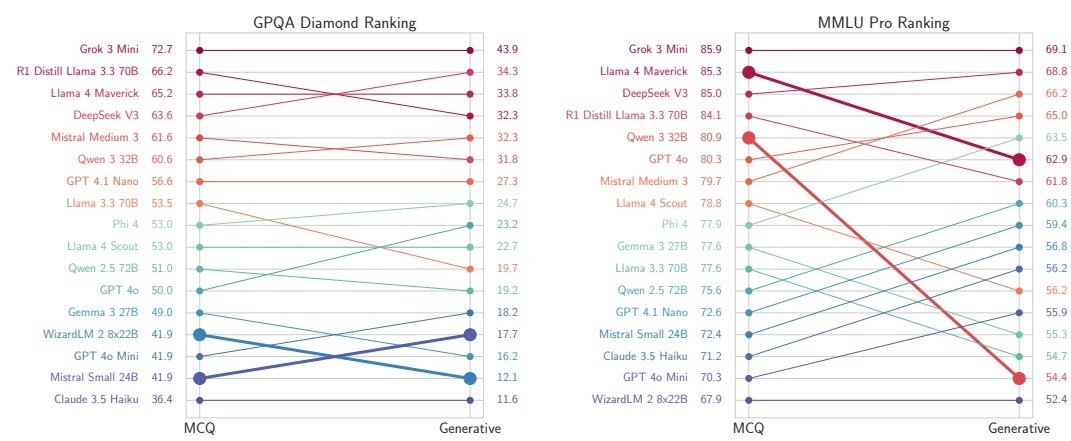

Figure 16: **Leaderboard rankings change on the whole unfiltered dataset** when moving from MCQ to answer-matching on generative responses in GPQA-Diamond (L) and MMLU-Pro (R): Thick lines represent statistically significant changers based on the Compact Letter Display algorithm (Piepho, 2004). We see chat-optimised proprietary models (GPT 4.1 Nano, 4o Mini, Claude 3.5 Haiku) climb on generative rankings, whereas open-weight models judged by their multiple-choice benchmark performance can (WizardLM 2, Llama 4 Maverick, Qwen 3 32B) drop markedly. The figure highlights that benchmark conclusions — and hence model selection — depend critically on the choice of evaluation protocol.

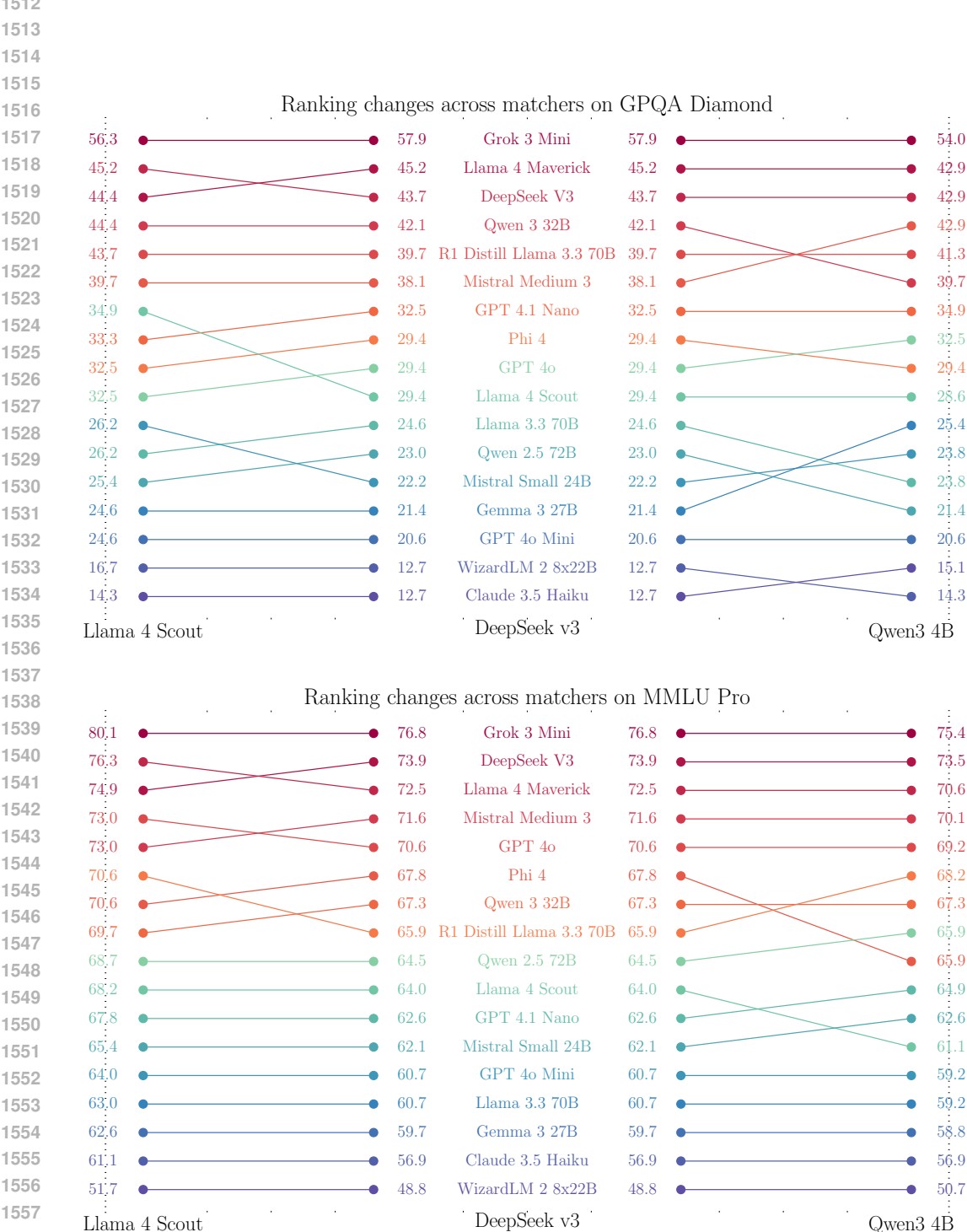

Figure 17: Ranking changes across matchers on GPQA-Diamond (left) and MMLU Pro (right) when the language model used as a matcher is varied. We find that there are only few ranking changes between matchers and none of the changes are significant.

# E    ERROR ANALYSIS OF ANSWER MATCHING MODELS

In addition to reporting aggregate accuracy, we perform a detailed error analysis of our answer matcher models. We focus on two representative systems: Qwen3-4B (a smaller matcher) and Llama-4-Scout (the strongest model in our evaluations, see Fig. 5). We analyze their disagreement cases on both MMLU PRO and GPQA DIAMOND, with the goal of identifying systematic failure modes.

**Error Categories.**    We find that matcher errors can be grouped into a set of recurring categories. Table 2 summarizes these categories, with representative examples drawn from Qwen3-4B and Llama-4-Scout outputs. A substantial fraction of errors are due to rigid adherence to format, units, or precision thresholds, even when the substantive content of the answer is correct. Other cases involve failure to recognize equivalent units, inconsistency in the required level of completeness, or genuine conceptual mistakes.

The examples illustrate that matchers often emphasize surface-level correctness (e.g., units, precision, formatting) over semantic correctness. Qwen3-4B, in particular, tends to reject valid answers that are phrased differently or use alternative units, while sometimes accepting semantically incorrect answers when they match in form. Llama-4-Scout, by contrast, shows more tolerance to format variation but can be overly permissive, occasionally accepting incorrect completions.

**Human Alignment Ceiling.**    Even with careful model design, it is important to note that inter-human agreement itself is not perfect. We observe Scott's $\kappa$ for human-human alignment at $0.87$ on MMLU PRO and $0.96$ on GPQA DIAMOND (see Fig. 5). Since human annotators can disagree or make mistakes, we do not expect current models to surpass these levels of agreement.

**Model-Level Disagreement Patterns.**    To quantify how strictness or leniency manifests in practice, we summarize the disagreement statistics for Qwen3-4B and Llama-4-Scout in Table 3. Qwen3-4B exhibits a bias towards false negatives, being overly strict in rejecting plausible answers. Conversely, Llama-4-Scout is biased towards false positives, reflecting its more lenient matching criterion.

**Summary.**    This analysis highlights that the main challenge for matcher models is balancing strictness and leniency: small models like Qwen3-4B tend to be too rigid, while stronger models like Llama-4-Scout lean towards permissiveness. Future work may focus on adaptive thresholds, semantic equivalence detection (e.g., unit conversions), and incorporating domain knowledge to reduce both false positives and false negatives.

| Root Cause | % of Disagreements | Qwen3-4B Example | Llama-4-Scout Example | Matcher (Qwen3-4B) Reasoning |
|---|---|---|---|---|
| Unit/Format Rigidity | 38% | Q: "What is the percentage of angular magnification" Target: `0.5%` Response: `0.5` Issue: Missing % symbol (FN) | Q: "What are the rabbinical commentaries produced after the Mishnah called?" Target: `Gemarah` Response: `Gemara` Issue: spelling variant rejected | "Response must have at least as much information as ground-truth; unit is essential" |
| Precision Over-Strictness | 23% | Q: "Time for stone to strike water" Target: `4 s` Response: `4.04 s` Issue: 1% error rejected (FN) | Q: "Find thermal conductivity" Target: $2.47 \times 10^{-4}$ Response: `6.27e-4` Issue: 87% error accepted (FP) | "Relative error greater than 1% threshold" |
| Unit Conversion Blindness | 15% | Q: "Calculate enthalpy" Target: `-2.72 kcal` Response: `-11.42 kJ` Issue: Equivalent units rejected (FN) | Q: "tRNA anticodon sequence" Target: $5'$-C-A-U-$3'$ Response: $3'$-G-U-A-$5'$ Issue: Wrong sequence accepted (FP) | "Different units of energy; values not equivalent" |
| Completeness Inconsistency | 13% | Q: "Geometrical position in qubit space" Target: `r=(0,0,0)` Response: `Origin` Issue: Correct but less specific (FN) | Q: "IUPAC nomenclature" Target: `3-bromo-4'-methoxy-1,1'-biphenyl` Response: `1-bromo-3-(4-methoxyphenyl)benzene` Issue: Different structure accepted (FP) | "Response lacks sufficient information detail" |
| Conceptual Errors | 11% | Q: "Color of light absorbed" Target: `Red` Response: `Green` Issue: Complementary color confusion (FP) | Q: "Stars detectable by both observatories" Target: `Star3, Star5` Response: `Star1, Star3, Star5` Issue: Extra incorrect star (FP) | Domain knowledge gaps in evaluation |

Table 2: Error categories and representative examples for Qwen3-4B and Llama-4-Scout. FN = False Negative, FP = False Positive.

| Model | Total Disagreements | False Negatives | False Positives | Primary Bias |
|---|---|---|---|---|
| Qwen3-4B | 51 | 37 (73%) | 14 (27%) | Overly Strict |
| Llama-4-Scout | 33 | 14 (42%) | 19 (58%) | Overly Lenient |

Table 3: Error distribution of matcher models compared to human consensus across MMLU PRO and GPQA DIAMOND.

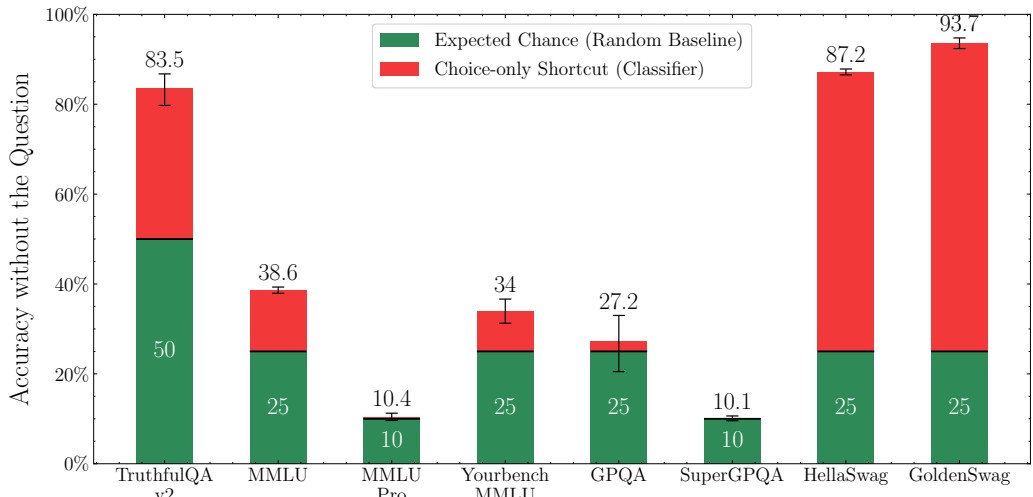

Figure 18: Shortcut accuracy achieved by finetuning DeBerta that sees only the answer choices, **without any access to the question**. We show the improvement over the random-guess baseline in red. In some datasets like TruthfulQA-v2, HellaSwag and MMLU, this small and old BERT-style model also achieves similar accuracy to the ones reported in Fig 3. However, we found that the model is unable to fit on benchmarks with > 4 options leading to baseline (constant choice) accuracy in MMLU-Pro and SuperGPQA.

## F    FURTHER DEMONSTRATING DISCRIMINATIVE SHORTCUTS IN MCQs

In Section 2, we showed that on popular multiple-choice benchmarks, fine-tuning Qwen3-4B model on choices-only prompt can lead to significant accuracy on the held-out test set. Here, we show that non-trivial shortcut accuracy can be achieved even by finetuning small embedding models like DeBERTa. More specifically, we finetune embeddings of the DeBERTa-v3-large model[2] (with roughly 500M parameters) and find that it achieves similar accuracy on TruthfulQA v2, MMLU and HellaSwag (Fig. 18). However, on benchmarks with 10 choices like MMLU Pro and SuperGPQA, the training loss doesn't fall showing DeBERTa is unable to bind prompts with high number of choices to their labels. It is nonetheless interesting that even a small, old BERT-style classifier can achieve non-trivial accuracy on some of the datasets.

Using the classifier, we find questions which it gets right (without choices) and we provide below examples of some questions from MMLU Pro where clearly choices-only shortcut exist[3]. These examples are only meant to show some ways in which a model may exploit shortcuts from just the choices to arrive at the correct answer. These examples are not exhaustive though.

---

[2]https://huggingface.co/microsoft/deberta-v3-large

[3]For MMLU Pro, we used Qwen3-4B as the classifier as it got signifcant accuracy.

---

**Biology MCQ Example — Question ID: 2842**

**Question: What cell components typically found in eukaryotic cells are missing from a bacterial (prokaryotic) cell?**

- A. Vacuoles, cytoskeleton, fimbriae
- B. Cellulose in the cell wall, thylakoids, glyoxysomes
- C. Flagella, pili, plasmids
- D. Centrioles, lysosomes, microtubules
- E. Nuclear membrane, histones, mitotic spindle apparatus, Golgi apparati, mitochondria, and endoplasmic reticulum
- F. Cell wall, ribosomes, plasma membrane
- G. Chlorophyll, intra-cellular structures, respiratory enzymes
- H. Nuclear pore complex, peroxisomes, chloroplasts
- I. Peptidoglycan layer, nucleoid region, capsule
- J. Cytoplasm, bacterial DNA, mesosomes

**Answer:** E

**Choices-only Shortcut:** Choice E is considerably longer than other choices and has many more components listed out in comparison.

---

**Chemistry MCQ Example — Question ID: 4525**

**Question: Assume that all gases are perfect and that data refer to 298.15 K unless otherwise stated. Calculate the total change in entropy, when a sample of nitrogen gas of mass** $14\,\mathrm{g}$ **at** $298\,\mathrm{K}$ **and** $1.00\,\mathrm{bar}$ **doubles its volume in an isothermal reversible expansion.**

- A. -4.8 J/K
- B. 2.3 J/K
- C. -1.0 J/K
- D. 1.5 J/K
- E. 3.2 J/K
- F. -2.3 J/K
- G. 0
- H. 5.0 J/K
- I. 4.1 J/K
- J. -3.5 J/K

**Answer:** G

**Choices-only Shortcut:** Choice G is the only choice with no unit listed whereas all other choices have 'J/K' listed.

# G  MODELS' RESPONSES IN MCQ VS FREE FORM

In Section 4, we discussed the cost of running a leaderboard using LM-based matcher for different models and compared it with the cost of multiple choice evaluations. Then, we briefly mentioned how the output tokens of a model can be much less in free-form generation (when the question is asked without choices) compared to multiple choice evaluations (Fig. 7). Here, in Fig. 21 we show that this trend is not specific to any one model but is true for all the models we evaluated, on both our datasets. We show below an example of this in practice on a mechanical engineering question from MMLU-Pro.

---

### Mechanical Engineering Free-Form Example — Question ID: 11475

**Question: Assume that all gases are perfect and that data refer to 298.15 K unless otherwise stated. Calculate the total change in entropy, when a sample of nitrogen gas of mass** $14\,\mathrm{g}$ **at** $298\,\mathrm{K}$ **and** $1.00\,\mathrm{bar}$ **doubles its volume in an isothermal reversible expansion.**

**Answer:** 0 J/K

**Model (Gemma3-27B-Instruct)'s Response:** Here's how to calculate the total change in entropy for the isothermal reversible expansion of nitrogen gas:

**1. Understand the Process**

- **Isothermal:** Temperature remains constant at 298.15 K.
- **Reversible:** Expansion is infinitesimally slow, maintaining equilibrium.
- **Nitrogen Gas:** Treated as an ideal gas.
- **Doubling Volume:** Final volume $V_2 = 2V_1$.

**2. Formula for Isothermal Reversible Expansion**
The entropy change is given by:

$$\Delta S = nR \ln \left( \frac{V_2}{V_1} \right)$$

Where:

- $n$ = number of moles of gas
- $R$ = ideal gas constant = $8.314\,\mathrm{J/mol \cdot K}$
- $V_2$, $V_1$ = final and initial volumes, respectively

**3. Calculate the Number of Moles** ($n$)

- Molar mass of $N_2$ = $28.0134\,\mathrm{g/mol}$
- Mass = 14 g
- 

$$n = \frac{14\,\mathrm{g}}{28.0134\,\mathrm{g/mol}} \approx 0.5\,\mathrm{mol}$$

**4. Apply the Formula**
Since $V_2/V_1 = 2$:

$$\Delta S = (0.5\,\mathrm{mol}) \cdot (8.314\,\mathrm{J/mol \cdot K}) \cdot \ln(2)$$
$$\Delta S = 0.5 \cdot 8.314 \cdot 0.6931 \approx 2.887\,\mathrm{J/K}$$

**Answer:** The total change in entropy is approximately:

$$\boxed{2.89\,\mathrm{J/K}}$$

---

1890
1891
1892
1893
1894
1895
1896
1897
1898
1899
1900
1901
1902
1903
1904
1905
1906
1907
1908
1909
1910
1911
1912
1913
1914
1915
1916
1917
1918
1919
1920
1921
1922
1923
1924
1925
1926
1927
1928
1929
1930
1931
1932
1933
1934
1935
1936
1937
1938
1939
1940
1941
1942
1943

## Mechanical Engineering MCQ Example — Question ID: 11475

**Question: Assume that all gases are perfect and that data refer to 298.15 K unless otherwise stated. Calculate the total change in entropy, when a sample of nitrogen gas of mass** $14 \text{ g}$ **at** $298 \text{ K}$ **and** $1.00 \text{ bar}$ **doubles its volume in an isothermal reversible expansion.**

- A. -4.8 J/K
- B. 2.3 J/K
- C. -1.0 J/K
- D. 1.5 J/K
- E. 3.2 J/K
- F. -2.3 J/K
- G. 0
- H. 5.0 J/K
- I. 4.1 J/K
- J. -3.5 J/K

**Answer:** G

**Model (Gemma3-27B-Instruct)'s Response:** Here's how to calculate the change in entropy for the isothermal reversible expansion of nitrogen gas:

**1. Understand the Process**

- **Isothermal:** Temperature remains constant at $298.15 \text{ K}$.
- **Reversible:** Expansion occurs slowly enough to maintain equilibrium.
- **Expansion:** Gas volume increases during the process.

**2. Formula for Isothermal Reversible Expansion**

$$\Delta S = nR \ln \left( \frac{V_2}{V_1} \right)$$

Where:

- $n$ = number of moles of gas
- $R$ = ideal gas constant = $8.314 \text{ J}/(\text{mol} \cdot \text{K})$
- $V_2$ = final volume
- $V_1$ = initial volume

**3. Calculate the Number of Moles ($n$)**

- Molar mass of $N_2 = 28 \text{ g/mol}$
- Mass of gas = 14 g
- 

$$n = \frac{14 \text{ g}}{28 \text{ g/mol}} = 0.5 \text{ mol}$$

**4. Determine the Volume Ratio**
The volume doubles, so:

$$\frac{V_2}{V_1} = 2$$

**5. Calculate the Entropy Change**

$$\Delta S = nR \ln \left( \frac{V_2}{V_1} \right) = (0.5)(8.314) \ln(2)$$

$$\Delta S = 0.5 \cdot 8.314 \cdot 0.693 \approx 2.88 \text{ J/K}$$

**6. Choose the Closest Answer**
The closest multiple-choice option to 2.88 J/K is:

E. 3.2 J/K

**Therefore, the answer is E.**

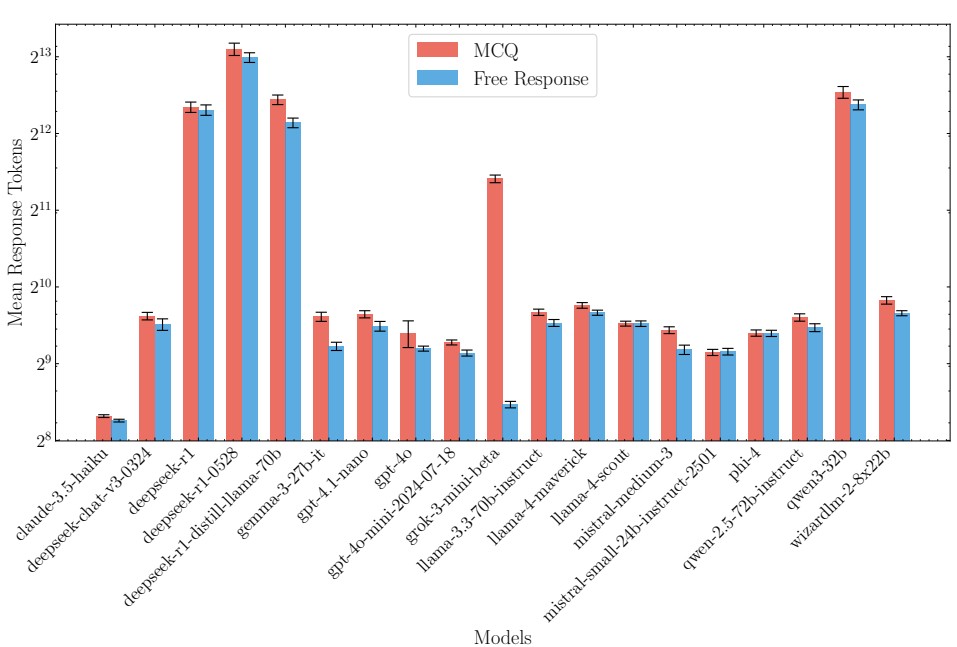

Figure 19: Mean output tokens on GPQA Diamond.

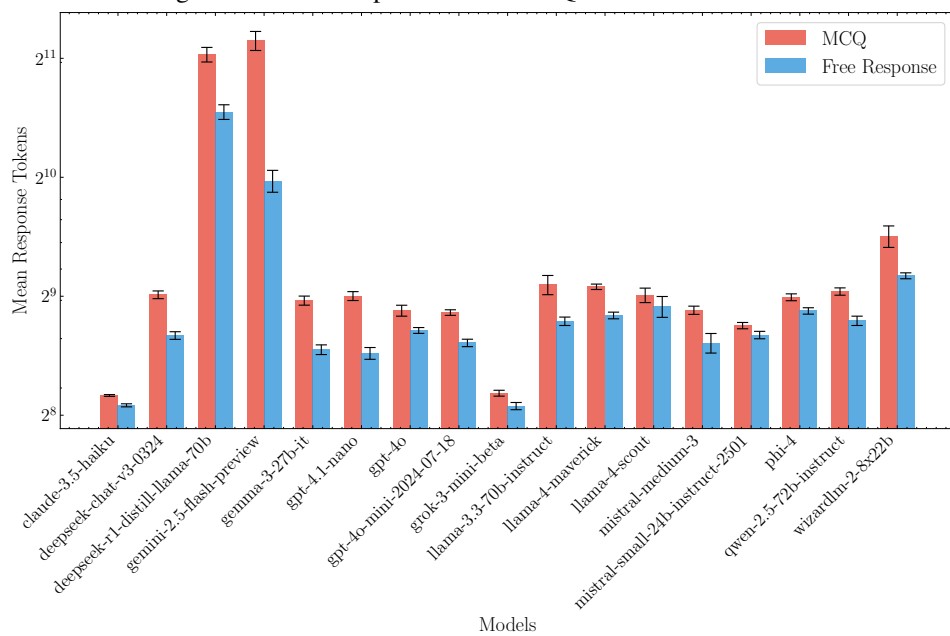

Figure 20: Mean output tokens on MMLU Pro.

Figure 21: **Comparison of the number of output tokens in MCQ vs Free Form evaluation** on GPQA-Diamond and MMLU-Pro. Each panel shows a bar plot on the mean output tokens produced by the models evaluated. We consistently find that the number of output tokens in free form (blue) is **lower** than multiple choice (red) *across all the models* evaluated.

# H    PROMPT TEMPLATES

## H.1    PROMPT FOR GROUND TRUTH MATCHING

**Prompt for matching responses with ground truth (in Python)**

```python
def get_judge_prompt_with_gt(question, target, response,
                             incorrect_options=None,
                             cot=True):
    """
    Generate a prompt for the judge with ground truth.

    Args:
        question: The question being asked
        target: The ground truth answer
        response: The response to judge
        incorrect_options: Optional string containing
            incorrect options
        cot: Whether to use a COT prompt

    Returns:
        A formatted prompt string for the judge
    """
    # The response can have more information than the
        ground-truth.
    # It can be more specific (for example, "Labrador" is
        more
    # specific than "dog"), or have additional possible
        answers.
    # But it must cover everything mentioned in the
        ground-truth.
    # It is okay if it covers it in different words, i.e.
        paraphrased.

    prompt = f"""Your task is to judge whether the given
        response to a question matches a given ground
        truth answer or not. You are provided with a
        question, a ground truth response, and the
        response you need to judge.
For a response to "match", it must have at least as much
    information as the ground-truth.
The response can have more information than the
    ground-truth. It can be more specific (for example,
    "Labrador" is more specific than "dog"), or have
    additional possible correct answers. But it must
    cover everything mentioned in the ground-truth. It is
    okay if it covers it in different words, i.e.
    paraphrased.
For numeric answers, the relative error, defined as
    |response - ground truth| / mean(response, ground
    truth), must be less than 1% for the response to be
    judged as a correct match. Here, if the ground truth
    is a specific numeric quantity but the response is a
    range, then they don't match (even if the range
    contains the ground truth).

Possible judgments:
```

```
    "0": The response does not match the ground-truth answer.
    "1": The response matches the ground-truth.

    Question: "{question}"
    Ground truth: "{target}"
    """

        if incorrect_options:
            prompt += f"\n{incorrect_options}"

        prompt += f"""Response: "{response}"

    Your job is to ONLY check whether the given response matches
        ↪ the ground truth answer or not in the context of the
        ↪ question. You DO NOT NEED to assess the correctness
        ↪ of the response. This is part of an automated
        ↪ evaluation process, therefore you MUST OUTPUT your
        ↪ final answer as "0" or "1" in <answer> </answer>
        ↪ tags."""

        if cot:
            prompt += "\nThink step by step and end your
                ↪ response with " + \
                    "<answer>0</answer> OR <answer>1</answer>
                        ↪ TAGS."
        else :
            prompt += "\nYOU SHOULD ALWAYS END YOUR RESPONSE
                ↪ WITH " + \
                    "<answer>0</answer> OR <answer>1</answer>
                        ↪ TAGS."

    # Think step by step and end your response with
    # <answer>0</answer> OR <answer>1</answer> TAGS.
    # YOU SHOULD ALWAYS END YOUR RESPONSE WITH
    # <answer>0</answer> OR <answer>1</answer> TAGS.

        return prompt
```

## H.2 PROMPT FOR LLM-AS-A-JUDGE WITHOUT GROUND TRUTH

**Prompt for LLM Judges checking correctness (without access to ground truth) in Python**

```
    def get_free_judge_prompt(question, response, cot=True):
        prompt = f"""Your task is to judge whether the given
            ↪ response to a question is correct or not. You are
            ↪ given a question and the response you are judging.
    Possible judgments:
    "0": The response is incorrect.
    "1": The response is correct.

    Question: "{question}"
    Response: "{response}"
```

```
The response should fully answer the question and must not
    ↪ be vague.
For numeric answers, the relative error, defined as
    ↪ |response - ground truth| / mean(response, ground
    ↪ truth), must be less than 1% for the response to be
    ↪ judged as a correct match. Here, if the ground truth
    ↪ is a specific numeric quantity but the response is a
    ↪ range, then they don't match (even if the range
    ↪ contains the ground truth).

To the best of your knowledge: Does the provided response
    ↪ answer the question correctly? This is part of an
    ↪ automated evaluation process, therefore you MUST
    ↪ OUTPUT your final answer as "0" or "1" in <answer>
    ↪ </answer> tags."""
    if cot:
        prompt += "\nThink step by step and end your
            ↪ response with " + \
                "<answer>0</answer> OR <answer>1</answer>
                    ↪ TAGS."
    else:
        prompt += "\nYOU SHOULD ALWAYS END YOUR RESPONSE
            ↪ WITH " + \
                "<answer>0</answer> OR <answer>1</answer>
                    ↪ TAGS."

    return prompt
```

