# OpenReview forum: "Answer matching outperforms multiple choice for LLM evaluations"
_ICLR.cc/2026/Conference — Submitted to ICLR 2026_

### Official Review · Reviewer_vg9i · 2025-10-29

**Soundness:** 3
**Presentation:** 4
**Contribution:** 4
**Rating:** 8
**Confidence:** 3

**Summary:**

This paper formalizes the use of LLMs in evaluation and benchmarking as a substitute for multiple choice questions. They propose to use an LLM-matcher, which is an LLM judge that has access to the reference answer (correct option per se). They show that MCQ measures a discriminative task and is vulnerable to "choice-only" shortcuts by fine-tuning a model on options only. They show that using "answer matching" and evaluating generations is well-aligned with human graders, is surprisingly cheaper in practice than MCQ, and reveals that SOTA models fall short in free-form generation on saturated MCQ benchmarks.

**Strengths:**

1. The paper is well-written, making all claims clear and supporting them with justifications and experimental results.
2. The work shows why MCQ is not useful for benchmarking LLMs by showing that they have discriminative shortcuts by fine-tuning a choice-only model and showing its surprisingly high performance on TruthfulQA.
3. They clearly describe how they go from MCQ to answer matching and how it's different than LLM Judges. Importantly, the results on MATH (which can even be automatically verified) shows near-perfect alignment where MCQ itself has a much lower agreement. The same results hold for GPQA and MMLU-Pro.
4. This work will have impact on benchmarking and leaderboard efforts. The authors show how using answer matching will change the ranking of models (Fig. 6) and surprisingly, using answer matching is cheaper than MCQ because of an interesting phenomenon where models generate longer responses when having access to options. They also discuss reliability and robustness of reusing MCQ benchmarks with answer matching.

**Weaknesses:**

1. The paper suggests "using *smaller* models" (L. 80) or "using *recent* models" (L 470) as the matcher but provides no scaling study. The paper would benefit from a scaling study that analyses how llm matcher size, release date, and cost affect performance in terms of alignment with human graders and clarify the optimal trade-off.

2. Using answer matching might not be feasible for all tasks or benchmarks (e.g., "which of the following options ...?" The authors also had to filter some MMLU-pro and GPQA questions manually to ensure there's a unique answer and the question can be answered without having access to questions. Adding a discussion on what kind of tasks can be restructured to be suitable for "answer matching" instead of filtering them would be beneficial in future benchmark creation efforts.

**Questions:**

1. Do you think answer matching can be used for tasks/question that don't have a ground-truth final answer (e.g., "List the main causes of climate change.")? What if reference responses are available? Does this mean the matcher needs to have access to all possible acceptable responses and in that case using llm-judge would be more feasible?

2. L 115: "specific enough that the correct choice can be uniquely inferred." How do you define "specific enough"?

---

> ### Author Response · Authors · 2025-11-17
>
> Thank you for your support of our work! We are happy to discuss the questions you asked.
>
> > The paper would benefit from a scaling study that analyses how llm matcher size, release date, and cost affect performance in terms of alignment with human graders
> >
>
> Thanks for the suggestion! We now add this for the Qwen3 and Gemma3 model families which provide various sizes in Figure 12 (Appendix E.1). For the Qwen3 family we find that on both MMLU Pro and GPQA Diamond, there is a rapid improvement from 1B to 4B, and after that the alignment plateaus, suggesting Qwen3 4B might be a sweet spot for cost-accuracy tradeoff. For the Gemma3 family, on GPQA Diamond we see alignment improve till 12B but not much after that, whereas for MMLU Pro 27B clearly is a better grader than 12B, perhaps because the latter is a more subjective dataset. This shows that the optimal choice of matcher size can be task dependent, and also depends on the cost-accuracy preferences of the user. That said, Qwen3 4B stands out as a cheap yet strong grader for both MMLU Pro and GPQA Diamond.
>
> > Do you think answer matching can be used for tasks/question that don't have a ground-truth final answer (e.g., "List the main causes of climate change.")?
> >
>
> Answer matching cannot be used for tasks/questions which don’t have ground truth as our definition of answer matching relies on a reference (correct) answer. We acknowledge this limitation in the limitations section in Appendix A (answer matching is hard if many answers are possible).
>
> > What if reference responses are available?
> >
>
> If reference responses are available, then answer matching can be used such that a response is marked correct if it matches of any of the available reference answers.
>
> > Does this mean the matcher needs to have access to all possible acceptable responses and in that case using llm-judge would be more feasible?
> >
>
> Ideally, yes. The more the number of (distinct) references answer available, the less is the chance of a correct answer being marked wrong. There is indeed a tradeoff where as more answers become possible, it may be beneficial to use LLM judges with rubrics instead.
>
> > Adding a discussion on what kind of tasks can be restructured to be suitable for "answer matching" instead of filtering them would be beneficial in future benchmark creation efforts.
> >
>
> We do provide a detailed discussion of this in Appendix Section B (Limitations and Considerations).
>
> > L 115: "specific enough that the correct choice can be uniquely inferred." How do you define "specific enough"?
> >
>
> We call a question/answer pair specific enough if the provided answer is the sole, unique correct answer to the provided question (without any choices). We clarify this below with an example. Consider the two questions below in mcq format:
>
> Q1: What is a prime number less than 10?
>
> A. 6  B. 8  C. 7  D. 9
>
> Correction Option: C
>
> Correct Answers: 2, 3, 5, 7
>
> Q2: What is the chemical symbol for gold?
>
> A. Ag  B. Au  C. Ga  D. Gd
>
> Correct Option: B
>
> Correct Answer(s): Au
>
> In Q1, if we remove choices, the correct answer cannot be *uniquely* inferred thus making it unsuitable for generative evaluation if we don’t have list of all correct answers (which we generally don’t have for MCQ benchmarks). However, in Q2, the correct can still be uniquely inferred given just the question so it is deemed specific enough in our annotation/filtering. Please see Table 1 from [1] for a more detailed breakdown of possible failure modes of strict match, some of which also apply to LLM-based answer matching.
>
> We hope that clarifies your questions, and are happy to discuss further.
>
> [1] Mañas, Oscar, Benno Krojer, and Aishwarya Agrawal. "Improving automatic vqa evaluation using large language models." *Proceedings of the AAAI Conference on Artificial Intelligence*. Vol. 38. No. 5. 2024.

---

> > ### Author Response · Authors · 2025-11-26
> > **Any updates after our response?**
> >
> > Hey,
> >
> > We responded to your questions a week back, and incorporated your suggestion of adding a scaling study for matchers. We want to check, are there any remaining questions, or did we address them satisfactorily?

---

### Official Review · Reviewer_DXCf · 2025-10-30

**Soundness:** 3
**Presentation:** 4
**Contribution:** 2
**Rating:** 4
**Confidence:** 4

**Summary:**

The paper discusses the shortcomings of MCQ benchmarks and demonstrates they can be prone to model shortcuts. They show that answer-matching (llm-as-a-judge with a reference answer) is a better evaluation that is more aligned with ground-truth and human evals.

**Strengths:**

The paper presents a systematic study aiming to show the weaknesses of MCQ benchmarks and why answer matching is a better alternative. They conduct grounded experiments to support their claims.

**Weaknesses:**

While the paper has some interesting experiments/findings (e.g. models can solve MCQ w/o conditioning on the question, answer matching is better aligned with ground-truth evals/human evals, model rankings change between MCQ and answer matching), the formulations of some of these experiments feel too weak to make definitive claims (see questions) and the analysis and intuitions around the findings feel a bit lacking. Overall, it is unclear if the paper is contributing knowledge beyond what is already known to the community – it is well known that discriminative tasks such as MCQ are easier than generative tasks (generator-verifier gap), LLM-as-a-judge with a reference answer (“answer matching”) is also a well known evaluation technique in the community and it is not surprising that creating generative versions of MCQ benchmarks would make these benchmarks harder.

**Questions:**

- Section 2:
Can multiple choice evals be answered w/o the question: if the MCQ benchmark is very in-distribution for the LLM (e.g. very similar to data it has been pre-trained on), the model could be internally “predicting” the question and then answering it, which means that by seeing similar training data it has learned a mapping between this type of answer choices and their associated questions and can infer the question from the answer choices. This doesn’t mean the model can isolate the correct answer just from the set of choices, just that it has memorized similar questions and answers from its data. Fine-tuning the model to select the correct answer choice makes this task even easier. A more convincing way to demonstrate that models can isolate the correct answer from a set of distractors would be taking an OOD MCQ benchmark and zero-shot prompting models to select the correct answer.
- Result 2: Why are model generated answer choices more prone to short cuts?
- Result 3: Which model is result 3 trained on?
-Discussion: While it is interesting that models can be trained to select the correct answer from a set of distractors w/o seeing the question, the key argument this supports is that discriminative tasks are inherently easier than generative tasks? However this generator-discriminator gap is well-known in machine learning circles, so not sure if the findings are adding something beyond that.

- section 3.1: Other works such as https://arxiv.org/pdf/2207.05221 study the calibration of language models across different types of tasks and show that they are well-calibrated on multiple choice and true-or-false tasks. This suggests that framing self-evaluation as a multiple choice task may be effective. How does this relate to your findings?

- section 3.1: Another reason why multiple choice cloze may be a poor method is due to surface form competition: https://arxiv.org/pdf/2104.08315, discussing this would be useful

- section 3.2: The dataset size of questions (493, 126) which results are reported on is perhaps too small to make conclusive statements

- section 4: It is interesting that model rankings change when transitioning from MCQ to answer matching, but analysis and intuitions on why this is are lacking. If MCQ questions allow models to exploit discriminative shortcuts, one might expect models of all strengths to benefit from this and perhaps stronger models (which also perform well under answer matching) to exploit these short cuts even more and rank high on both types of benchmarks.

- section 4: How is answer matching cheaper than multiple choice evals? Answer matching would typically require an LLM call as multiple equivalent surface forms may exist, while multiple choice evals can be done w/ regex-based extractions and rule-based matching

---

> ### Author Response · Authors · 2025-11-17
>
> We are glad you found our study systematic and claims well grounded. We hope to address all your concerns below.
>
> ## Clarifying Contributions
>
> > it is unclear if the paper is contributing knowledge beyond what is already known to the community
> >
>
> We do not know of any work showing our specific contributions before:
>
> (1) MCQ benchmarks are a discriminative task, allowing LLMs to achieve high accuracies (Figure 3) on them even without showing the model the question (or image, in the case of MMMU Pro). Therefore, accuracy in these benchmarks may not be indicating the model’s ability to answer questions correctly in free-form format, which is a more realistic use-case for chatbot users.
>
> (2) Responses to free-form questions can now be evaluated reliably at scale at low costs (Figure 7), with small locally hostable models, that ensures easy reproducibility. While LLM based answer matching evaluations could be considered noisy at the outset, with a meticulous human annotation study we find they align far better with human grading than the multiple choice format (which has alignment equivalent to Llama 2 7B) for standard MCQ benchmarks like MMLU Pro and GPQA (Figure 5).
>
> The second point is not trivial. Until very recently, prominent papers have argued for the creation of more MCQ benchmarks as they considered LLM grading to be costly or not reproducible [1], as they thought it requires expensive proprietary models that may get depreciated to be a reliable alternative. We show that while this is true for LLM as a Judge without a reference answer, providing the reference answer simplifies the verification problem to semantic matching, making even small models much better at it.
>
> > generator-discriminator gap is well-known in machine learning circles
> >
>
> We agree. Note that our contribution lies in framing various shortcuts people have been noticing in MCQ benchmarks recently [2, 3] as arising from the discriminative nature of multiple choice. [2, 3] had tried to remove these issues by changing the MCQs, but we show that their updated datasets (Truthful QA v2, goldenswag) actually ended up worsening the problem by making the discriminative task easier, as we discuss in L185, 206. We hope that our work makes the community realize that the fix to shortcuts in multiple choice is not by removing by hand those that are easy to identify, but rather by moving away to generative evaluations, as LLM based answer matching has become a scalable superior alternative.
>
> > Other works such as https://arxiv.org/pdf/2207.05221 study the calibration of language models across different types of tasks and show that they are well-calibrated on multiple choice and true-or-false tasks. This suggests that framing self-evaluation as a multiple choice task may be effective. How does this relate to your findings?
> >
>
> We think calibration of language models is complementary to our work. Our contribution is to show that models can guess the correct answer to some samples without actually solving the question, inferring the answer from choice-only shortcuts. Work on calibration like the one linked relates the probabilities assigned by models over choices to their accuracy. They do not study **how** this accuracy or probability arise. It could be arising from both, actually solving the question, or merely inferring the answer using choice-only shortcuts.
>
> If the response above did not clarify your question, could you please clarify what you meant by “framing self-evaluation as a multiple-choice task”? We are not sure if we understood it correctly.
>
> > Another reason why multiple choice cloze may be a poor method is due to surface form competition: https://arxiv.org/pdf/2104.08315, discussing this would be useful
> >
>
> Thanks for the suggestion, we have incorporated this on `L310`.
>
> Continued below...

---

> > ### Author Response · Authors · 2025-11-17
> > **Response continued...**
> >
> > ## Methodology
> >
> > > This doesn’t mean the model can isolate the correct answer just from the set of choices, just that it has memorized similar questions and answers from its data. … A more convincing way to demonstrate that models can isolate the correct answer from a set of distractors would be taking an OOD MCQ benchmark and zero-shot prompting models to select the correct answer.
> > >
> >
> > While this is an interesting hypothesis, we believe our results on GoldenSwag and YourBench-MMLU provide evidence in favour of our hypothesis of choice-only shortcuts, rather than memorization. Both GoldenSwag and YourBench-MMLU were released in April 2025, after the release date of the Qwen3 model we used. On both these datasets, we find that the finetuned choice-only classifier attains higher accuracies (93% in GoldenSwag where chance accuracy is 25%!) where  than the original versions they “improve upon” which were released earlier (and more likely to be contaminated). Note that zero shot prompting scores are lower on the newer variants misleadingly indicating they are harder, when actually choice only shortcuts are easier to find in them with finetuning.
> >
> > There is a theoretical justification for using finetuning to surface choice only shortcuts. The statistical separability of choices (without the question) can be measured as the total variation distance. Any (trained) classifier lower-bounds the total variation distance, i.e. the separability. Thus, by training the classifier, we provide effectively a lower-bound on the choice-only theoretical separability of benchmark samples.
> >
> > We also encourage you to see the qualitative examples we present in Section G and H which provide clear evidence of choice-only shortcuts. We hope this clarifies your concerns. We are unsure what would constitute an OOD benchmark for pretrained language models, as they are trained on most of the internet. In fact, the training data of most popular language models is not publically released, so it is unclear how one can determine what is OOD. Does the reviewer have any particular benchmarks in mind? We would be happy to try our choice-only shortcuts experiment on these.
> >
> > > Why are model generated answer choices more prone to short cuts?
> > >
> >
> > This is an interesting question. We do not causally understand the mechanisms for why LLM generated MCQs have more shortcuts, and this is an interesting direction for future work. It is worth noting that the “human created” benchmarks like MMLU are sourced from real-world exams like standardized testing used to evaluate humans for high-stakes career affecting outcomes. It is likely that more care went into ensuring choice-only heuristics were less helpful for such high-stakes exams. GPQA was created by humans without being sourced from existing exams, and once again special care was taken to avoid choice-only shortcuts as we mention in `L199`, by finetuning a choice-only classifier and rejecting questions which it answered accurately. It is possible that such a step could help avoid this issue even in benchmarks with LLM generated questions, but we find at present this is a significant issue with existing benchmarks made using LLMs.
> >
> > > If MCQ questions allow models to exploit discriminative shortcuts, one might expect models of all strengths to benefit from this and perhaps stronger models (which also perform well under answer matching) to exploit these short cuts even more and rank high on both types of benchmarks.
> > >
> >
> > This is an interesting question that would require a deeper investigation.
> >
> > It is not necessary that stronger models exploit short cuts more. For example, parallel work [4] has shown that test time reasoners exploit choice only shortcuts only sometimes, and instead their accuracy increases more consistently when they are shown both the question and choices.
> >
> > In `L402-409` we discuss how anecdotally it seems models like the GPT family, offerred as chatbots on company APIs, seem to perform better on generative evaluations. This could be because other models like R1 distills, Gemma etc. are mostly judged based on their benchmark performance and not directly used for chat that often. This incentive could make model developers optimize (not necessarily on purpose, but perhaps implicitly, such as in checkpoint selection) for MCQ benchmarks in a way that does not align with real-world generative use.
> >
> > > How is answer matching cheaper than multiple choice evals?
> > >
> >
> > Evaluation costs consist of two components: the cost of generating outputs from the model, and the cost of running the evaluation rule. While the evaluation rule is indeed for cheap rule-based matching for MCQ, as we note in `L419-427`, models generate much more tokens when answering questions posed in multiple choice format compared to free-form (see Figure 7). This effect dominates the much smaller additional cost of the tokens generated by the answer matching model, even when using a large model like DeepSeek v3 for answer matching.

---

> ### Author Response · Authors · 2025-11-17
> **Response Conclusion**
>
> ### Final Questions
>
> > Which model is result 3 trained on?
> >
>
> Result 3, which shows that for the “multimodal benchmark” MMMU Pro questions can be answered **without the image or the question**, is based on finetuning the same Qwen3-4B language model backbone we used for the remaining results in Section 3.
>
> > The dataset size of questions (493, 126) which results are reported on is perhaps too small to make conclusive statements
> >
>
> We have included error bars based on bootstrapping in all our results. The error bars are indeed wider for GPQA (126 free form answerable questions). Nevertheless all our claims are statistically significant as evident from the plots. Are there any specific claims the reviewer is concerned about?
>
> ## Overall
> > the formulations of some of these experiments feel too weak to make definitive claims (see questions) and the analysis and intuitions around the findings feel a bit lacking.
> >
>
> We hope we have addressed all your questions about our experiments and contributions above. We are happy to discuss further! Overall, we show how popular LLM capability benchmarks suffer from discriminative shortcuts, and these can be identified by finetuning classifiers based on relatively small models. We show how the divergence between multiple choice discriminative evals and the generative use-case of language models leads to low alignment with human grading, demonstrating with a human annotation study how LLM based answer matching has recently become a cheap, scalable alternative. We believe these finds are valuable to the community, in that they may convince more benchmark developers to consider moving away from multiple choice evaluations.
>
> ### References
> [1] Automated Generation of Challenging Multiple-Choice Questions for Vision Language Model Evaluation. CVPR 2025, Zhang et al.
>
> [2] What the hellaswag? On the validity of common-sense reasoning benchmarks. arXiv:2504.07825, 2025, Chizhov et al.
>
> [3] New, improved multiple-choice truthfulqa. 2025, Evans et al.
>
> [4] Test-Time Reasoners Are Strategic Multiple-Choice Test-Takers. arXiv:2510.07761, 2025, Balepur et al.

---

> ### Author Response · Authors · 2025-11-26
> **Any updates after our response?**
>
> Hey,
>
> We posted our response to your review a week ago. We tried to answer all your questions, and clarify our contribution. Does this address your concerns? We would be grateful if you could let us know.

---

### Official Review · Reviewer_duJG · 2025-11-02

**Soundness:** 4
**Presentation:** 4
**Contribution:** 1
**Rating:** 4
**Confidence:** 4

**Summary:**

Compares three LLM-as-judge approaches, multiple choice vs. answering matching vs. reference-free evaluation. Under the multiple choice approach, the model under evaluation is given a question and a set of possible answers, judging is automatic from the answer key. Under the answer-matching approach, an LLM-as-judge is given a reference answer. In the final approach, the LLM-as-judge determines the correctness of the answer without a reference. While correct answers are unambiguous, they show that multiple choice allows models to take short-cuts in answering, reporting an interesting experiment in which models can successfully answer (with accuracy significantly greater than random) multiple choice questions from benchmarks with only the answers available. On the other hand, the use of reference answers can achieve accuracy similar to humans.

The paper is well written and organized. The topic is interesting. The experiments appear to be properly conducted and reported.

I have serious concerns about novelty, which I will detail in the "weakness" section.

**Strengths:**

The paper is well written and organized. The topic is interesting. The experiments appear to be properly conducted and reported.

I particularly enjoyed that "question free" multiple choice evaluation.

**Weaknesses:**

The abstract concludes, "In light of these findings, we discuss how to move the evaluation ecosystem from multiple choice to answer matching." This sentiment is repeated throughout the paper, including the conclusions. I am likely in a different part of the ecosystem (natural language question answer and RAG) but comparison against a reference answer already seems standard. I was trying to find the point where it became standard, and a brief Google search turns up papers like https://aclanthology.org/2021.mrqa-1.15/ and https://aclanthology.org/2023.acl-long.307/ that are not cited, plus others papers that are cited, but not fully differentiated from this work.

Here in 2025, I would view matching to a reference answer as a completely standard and uncontroversial approach to LLM evaluation.

**Questions:**

Is this not already standard?

---

> ### Author Response · Authors · 2025-11-17
> **Clarifying our contribution**
>
> We are glad you found our experiments interesting and properly conducted. Below, we hope to clarify the novelty of our contributions.
>
> > I am likely in a different part of the ecosystem (natural language question answer and RAG) but comparison against a reference answer already seems standard. I was trying to find the point where it became standard, and a brief Google search turns up papers like https://aclanthology.org/2021.mrqa-1.15/ and https://aclanthology.org/2023.acl-long.307/
> >
>
> Thanks for pointing us to the use of BERT based semantic similarity classifiers to grade responses in the retrieval literature, proposed as alternatives to lexical matching. We have included this, and further expanded our related work, shifting it from Appendix A to the main paper (Section 5) to clarify our contribution. We discuss how our contribution is unique to the literature below.
>
> Popular benchmarks for LLM capabilities like MMLU Pro and GPQA, used across model releases by companies, use the multiple choice format. We show that the same datasets can be evaluated with LLM based answer matching to obtain evaluations with far better alignment with human evaluations. While we are not the first to propose LLM based answer matching as an evaluation strategy (as we acknowledge early, `L113`), we are the first (to the best of our knowledge) to show its superiority for evaluating models on the questions in standard multiple choice benchmarks (see Figure 4,5). Our discussion on shortcuts in these benchmarks (Figure 3) that you enjoyed is also unique, and builds motivation for the shortcomings of multiple choice.
>
> People continue to create MCQ benchmarks for numerous reasons like “cost” (of using proprietary models for reliable answer matching), “reproducibility” etc. [1]. We show that with recent advances in LLM capabilities, we can now use small, locally hostable open-weight language models for answer matching with high validity. As shown in Figure 7, empirically across 18 models we find that it is actually more costly to evaluate in multiple choice format than generative with answer matching. This is because of the new regime where models use varying test time compute depending on the task, and we find the answerer outputs extra tokens for MCQs than generative, which dominates the small additional cost of a language model performing the relatively simpler task of answer matching for grading.
>
> All of this is to say, we think there is immense scientific value in performing a head-to-head comparison between two fundamental evaluation paradigms, especially when the seemingly worse one (multiple choice) is actually more “standard” for LLM capability evaluations. We hope our work convinces more benchmark developers to consider switching from multiple choice to answer matching, and are happy to address any remaining concerns or questions you have about this.
>
> [1] Automated Generation of Challenging Multiple-Choice Questions for Vision Language Model Evaluation. CVPR 2025, Zhang et al.

---

> > ### Comment · Reviewer_duJG · 2025-11-24
> >
> > I've spent some time reading the other reviews and your responses. All the reviews raise the point that "answering matching" seems so well established that the stated focus of the paper seems misplaced.
> >
> > I share your concerns that "people continue to create MCQ benchmark" and that there is "scientific value in performing a head-to-head comparison," but I continue to wonder if your comparison adds enough to what is a well-understood problem.
> >
> > My initial rating was 4, and I continue to think that your paper is "marginally below the acceptance threshold. But would not mind if paper is accepted"

---

> ### Author Response · Authors · 2025-11-25
> **Our contribution is showing answer matching is a scalable alternative to MCQ, not the "problem" or the "answer matching" method itself**
>
> Thanks for clearly articulating your concern. This has helped us understand where the gap might be, so we want to try and bridge it.
>
> It is true that we are not the first to propose or use answer matching (`L113`). In that sense, yes, answer matching is a known method in the community. We also agree that problems with the multiple choice format have been highlighted before (`L530`). HOWEVER:
>
> Multiple choice has long been considered a necessary evil for LLM benchmarking, due to the lack of a scalable, reliable alternative. To the best of our knowledge, we are the first to show, that standard MCQ benchmarks are much better evaluated using answer matching, which presents a scalable alternative due to the recent improvement in language model capabilities (especially small, local ones). Further, we show many novel examples in Section 2 (MMMU-Pro, TruthfulQA v2, YourBench-MMLU, GoldenSwag) where the community tried to improve a benchmark, but these efforts were misplaced as they did not realise that the issue runs deeper: the fundamentally discriminative nature of multiple choice will always be prone to shortcuts, that can only be fixed with generative evaluations using answer matching.
>
> While our contribution is indeed nuanced, this should NOT be mistaken for it being small. Benchmarking is fundamental to Machine Learning progress, and we show the clear superiority of an emerging paradigm over the status quo for LLM evaluations.
>
> We hope you now appreciate the importance of our comparison in showing a clear path forward for a long-standing problem.

---

### Official Review · Reviewer_BPXQ · 2025-11-03

**Soundness:** 2
**Presentation:** 3
**Contribution:** 1
**Rating:** 2
**Confidence:** 4

**Summary:**

This paper discuss multiple-choice evaluation (MCQs) and generative evaluation in LLM evaluation.
The authors argue that MCQs inherently test discriminative rather than generative abilities and can be answered correctly even without reading the question, due to choice-only shortcuts.
They propose Answer Matching, a scalable generative evaluation method: a model generates a free-form answer given only the question, and a second “matcher” model determines if the response semantically matches a reference answer.
Through extensive experiments on MATH, MMLU-Pro, and GPQA-Diamond, the authors show that answer matching:
1. Achieves near-human agreement with ground-truth or human grading (Scott’s π ≈ 0.9–0.97),
2. Outperforms MCQ and LLM-as-a-judge (without reference answers),
3. Alters model rankings significantly,
4. Is no more costly—and sometimes cheaper—than MCQ evaluation.

The paper concludes that answer matching has recently become a viable, superior alternative for evaluating generative capabilities of LLMs.

**Strengths:**

1. The paper is well-written.

**Weaknesses:**

1. While we systematically evaluate the reliability of reference-guided matching (“Answer Matching”), similar judge-with-reference setups have already been widely adopted in practical benchmarks such as NovelQA (Wang et al., 2023), LongBench/LongBench-R (Bai et al., 2023/2024), RAGAs (Es et al., 2023) and MT-Bench (Zheng et al., 2023, LMSYS). These works all rely on an LLM to decide whether a generated answer semantically matches a gold reference, demonstrating that such paradigms are already standard practice in QA evaluation .

2. The core methodology of using LLM-as-a-judge with a reference answer to evaluate generative QA has already been explored in a number of studies.
For example, Evaluating OpenQA Evaluation (NeurIPS 2023) directly compared multiple-choice evaluation against generative answer judging and found that LLM judges better align with human graders. Similar comparisons appeared in Rethinking Evaluation of Open-Domain QA (Min et al., EMNLP 2021), Beyond Multiple Choice (Balepur et al., 2024), and JudgeBench (Tan et al., ICLR 2024).

**Questions:**

none

---

> ### Author Response · Authors · 2025-11-17
> **The relation of our work to the cited papers (when they exist)**
>
> Thanks for pointing us to NovelQA. We have added the following discussion about it to our related work:
>
> NovelQA uses both multiple choice and answer matching evaluations for a specific task, but does not directly compare their validity. In contrast, our contribution lies in showing that popular LLM benchmarks like MMLU-Pro and GPQA are better evaluated using answer matching than multiple choice due to the discriminative shortcuts in multiple choice.
>
> We are surprised by the following incorrect citation claims in this review:
>
> > …LongBench/LongBench-R (Bai et al., 2023/2024), RAGAs (Es et al., 2023) and MT-Bench (Zheng et al., 2023, LMSYS). These works all rely on an LLM to decide whether a generated answer semantically matches a gold reference
> >
>
> **None** of the papers mentioned above use LLM-based answer matching.
>
> LongBench uses Rouge and F1 score, not LLM based answer matching. We could not find any paper called “LongBench-R”. Did you mean LongBench-v2? If so, we once again could not find anything like LLM based answer matching.
>
> Ragas uses llm based evaluations but not matching to a reference answer.
>
> MT bench uses LLMs for preference evaluation, i.e. choosing between two LLM outputs, rather than evaluating a response by matching it with the reference answer.
>
> > For example, Evaluating OpenQA Evaluation (NeurIPS 2023) directly compared multiple-choice evaluation against generative answer judging and found that LLM judges better align with human graders. Similar comparisons appeared in Rethinking Evaluation of Open-Domain QA (Min et al., EMNLP 2021), Beyond Multiple Choice (Balepur et al., 2024)…
> >
>
> In “Evaluating OpenQA Evaluation (NeurIPS 2023)”, we did not find any mention of multiple choice evaluation.
>
> JudgeBench compares different LLM judges for preference evaluations. It neither includes MCQ evaluations, nor answer matching style evaluations using LLMs to compare to a reference.
>
> We also could not find the following papers you mentioned:
> 1. “Rethinking evaluation of Open-Domain QA” by Min et al.,
> 2. “Beyond multiple choice” by balepur et al.
> Could you please link us to the exact papers you meant so we can also discuss them?

---

> ### Comment · Reviewer_BPXQ · 2025-11-20
> **Apology for Misunderstanding and Notice of Review Revision**
>
> Thanks for the authors' response.
>
> Upon carefully reading the response and re-reading the paper, I realized that I had initially misunderstood the core concepts of the work. I initially interpreted this work as a methodological paper, which prompted my concern regarding the novelty of the approach (given that Answer Matching is standard practice). Following the authors' clarifications in the rebuttal, I now recognize the work's primary focus is analytical. Consequently, my previous summary and the weaknesses were incorrect. I sincerely apologize for this oversight.
>
> Thus, **I have updated a revised review.** Please check it.
>
> (Additionally, I would like to clarify a few points from my previous comments: "Longbench-R" was a typo and intended to refer to Longbench-v2. When I mentioned "Rethinking evaluation of Open-Domain QA," I was referring to the paper Evaluating Open-Domain Question Answering in the Era of Large Language Models; similarly, "Beyond Multiple Choice" referred to Which of These Best Describes Multiple Choice Evaluation with LLMs?. These errors occurred because I inadvertently used personal shorthand from my notes when compiling the review, and I apologize again for the confusion caused by my oversight. )

---

> ### Author Response · Authors · 2025-11-20
> **Author Response to New Review**
>
> Thank you for revising your review. We address your new concerns below:
>
> > The empirical validation relies heavily on reasoning-intensive datasets ... to robustly demonstrate the trade-offs between Answer Matching and MCQA, the evaluation should extend to a wider range of domains, such as Open-domain QA , Coding QA, Logical QA and Psychological QA.
>
> **We were unable to find any of these datasets.** Could you link us to which datasets you specifically mean? Note that we already show results on MMLU-Pro, which encompasses 14 broad categories including Psychology and Computer Science, while also testing logical reasoning. If your concern is our study is on reasoning-intensive datasets, we are not sure a) why that is a problem? b) would coding not be a reasoning task too? Note that it is time intensive to collect human grading annotations needed for our analysis, which is why we did them for MMLU-Pro and GPQA which are two of the most popular benchmarks for frontier LLM capabilities. Replicating this on more datasets would be difficult in the rebuttal period, but if you have in mind a really important benchmark that you think our results might not generalize to, and the community would be interested in, we are happy to try.
>
> > How does answer matching handle multi-reference or paraphrase-rich datasets (e.g., free-form QA, MRC)?
>
> When there are multiple reference answers possible, we suggest listing them all as part of the benchmark, see `L119-120` so the matcher can compare the response to all reference answers. If there are too many answers possible to enumerate (e.g. in coding), then we acknowledge in `L938-944` that answer matching is not the right evaluation.
>
> Answer matching is precisely motivated by its ability to handle paraphrases (in contrast to alternatives like exact match). We elaborate on this in `Appendix B.3 L1074-1080`. Particularly, paraphrase detection seems easier than verification, especially for hard reasoning problems.
>
> > Have you tested matcher robustness when reference answers are partially incorrect or ambiguous?
>
> If the dataset has incorrect/ambiguous reference answers, that is a dataset issue which is not the burden of an evaluation format (whether it be answer matching or MCQ).
>
> > While the paper positions the analysis of MCQ shortcomings (specifically regarding "shortcuts") as a primary contribution, this topic has been extensively explored in recent literature. Prior works such as [1], [2], and [3] have already highlighted the structural flaws and lack of robustness in MCQ evaluations. Notably, [4] has specifically addressed the "shortcut learning" phenomenon in QA models.
>
> Thanks for pointing us to [4], which we will cite and discuss in our next revision. This is an interesting paper showing some analysis of how shortcut learning occurs, and how it can be mitigated. We did indeed already cite some such works showing shortcuts in older NLP tasks in `L489`. We think our demonstrations of shortcuts on current frontier benchmarks still hold value, especially in highlighting how such shortcuts are a fundamental flaw of the discriminative nature of MCQs, motivating the switch to generative evals with answer matching.
>
> We extensively discussed the relation of our work to [1-3] in `L202-228`. While [1-3] are great papers showing shortcomings of MCQs by showing they are sometimes answerable with prompting without the question, our contribution is different. We show that choice-only prompting only reveals a symptom, but not the root cause of the problem, which we identify as multiple choice inherently being a discriminative task. Finetuning a classifier is a better way to reveal discriminative shortcuts, and is supported by theory, where any classifier lower-bounds the statistical separability (total variation distance) of classes. This crucial distinction is perhaps best illustrated by our discussion (`L202-211`) of Goldenswag, an update to Hellaswag designed so that prompting leads to lower accuracy. By finetuning a classifier, we reveal how the newer, "golden" version of this dataset actually suffers far more from choice-only shortcuts! We also show a similar results for TruthfulQA v2, which was derived from TruthfulQA (see `L182-190`). Moreover, we show this issue applies beyond language model benchmarks, even to MMMU-Pro, a "multimodal" benchmark where we can get high accuracies with the question or the image (`L212-215`).
>
> We hope these clarifications increase your support for our work.

---

> > ### Author Response · Authors · 2025-11-26
> >
> > Hey,
> >
> > We posted our response to your new review a week back, and were wondering if this addresses your remaining questions and concerns. If so, we would be grateful if you could let us know.

---

### Author Response · Authors · 2025-12-03
**Final author remarks**

We thank the reviewers for their time and feedback. 3/4 reviewers did not have time to acknowledge our response to their review, so we hope this short summary helps the new AC understand how we addressed all reviewer feedback.

All reviewers recognized that our *“claims are clearly supported*” with “*systematic”* experiments, the paper “*well-written”*, and were overall convinced by our main findings. The only concern was about novelty, for which we extensively discussed how our contribution is quite different from each of the papers mentioned by the reviewers. We also updated our PDF with an extensive Related Work section to clarify our contribution. Particularly, this is NOT a “methods” paper. As we explicitly acknowledge in the Introduction itself, our contribution is NOT “inventing” LLM based answer matching. Instead, we demonstrate how the status quo in LLM benchmarking, multiple choice questions, can be repurposed for generative evaluation via LLM based answer matching. By collecting human annotations on GPQA and MMLU Pro, we demonstrate how answer matching not only has superior alignment with human grading, but is surprisingly also more scalable and cheaper. We motivated this finding by highlighting how a finetuned classifier can obtain high accuracies across popular language (and multimodal) benchmarks, **without the question (or image)**, by exploiting discriminative shortcuts in multiple choice. As `Reviewer vg9i` emphasised, *“this work will have impact on benchmarking and leaderboard efforts.”*

One anomaly occurred during the rebuttal phase. `Reviewer BPXQ` completely changed their review (**increasing their score from 2 to 4**) and acknowledged that they had miscited papers in their initial review. We then also responded to suggestions in the new review. Overall, we answered all questions raised by the reviewers, and incorporated all suggestions to the PDF (changes colored in $\color{blue}{blue}$), including `Reviewer vg9i`'s request for a new experiment testing the impact of matcher size in Appendix D.1

---

### Meta-Review · Area_Chair_o3o6 · 2026-01-11

**Summary:**

This paper studies and analyzes the limitations of multiple choice question benchmarks, and discusses the importance of Answer Matching, a method where LLMs solve free-form problems and their answers are evaluated by another LLM. Through various experiments, the authors demonstrate the limitations of multiple choice questions and show that the answer matching is a better evaluation method (e.g., MMLU-Pro, GPQA, etc.).

A common concern shared by the AC and reviewers is the novelty of the method. At this point, the answer matching strategy is already a common and standard evaluation approach. While demonstrating the limitations of multiple choice benchmarks and showing that rankings can change under Answer Matching is meaningful, as the authors emphasize, the AC believes this falls below the acceptance threshold when considering the significance expected at an ICLR venue. Therefore, the AC recommend to reject this paper.

**Reviewer Concerns:**

The concerns regarding novelty and significance raised by Reviewer duJG and Reviewer DXCf remain unresolved. As of now, the answer matching strategy has become a well-established and widely adopted in real-world. While the rebuttal addressed this concern to some extent, the AC (and reviewers) still finds that the paper does not meet the acceptance threshold given the level of significance expected at an ICLR venue.

**Reviewer Scores:**

- Reviewer BPXQ: 2->4 as mentioned in the discussion.
- Reviewer duJG: 4->4. Given the critical concern about the paper's novelty and significance, the score has not changed.
- Reviewer DXCf: 4->4. Given the critical concern about the paper's novelty and significance, the score has not changed.
- Reviewer vg9i: 8->8 (unchanged).

---

### Decision · Program_Chairs · 2026-01-26

Reject